# Live-cell RNA imaging with the inactivated endonuclease Csy4 enables new insights into plant virus transport through plasmodesmata

David Burnett[1,2☯], Mohamed Hussein[1,2,3☯], Zoe Kathleen Barr[1,2¤a], Laura Newsha Näther[1¤b], Kathryn M. Wright[2], Jens Tilsner[1,2]*

1 Biomedical Sciences Research Complex, The University of St Andrews, School of Biology, St Andrews, Fife, United Kingdom, 2 Cell and Molecular Sciences, The James Hutton Institute, Invergowrie, Dundee, United Kingdom, 3 Cukurova University, Institute of Natural and Applied Sciences, Saricam, Adana, Turkey

☯ These authors contributed equally to this work.
¤a current address: Institute of Developmental Genetics, Heinrich-Heine University, Düsseldorf, Germany
¤b current address: Institute for Virology, Universitätsklinikum Freiburg, Freiburg, Germany
* jt58@st-andrews.ac.uk

## Abstract

Plant-infecting viruses spread through their hosts by transporting their infectious genomes through intercellular nano-channels called plasmodesmata. This process is mediated by virus-encoded movement proteins. Whilst the sub-cellular localisations of movement proteins have been intensively studied, live-cell RNA imaging systems have so far not been able to detect viral genomes inside the plasmodesmata. Here, we describe a highly sensitive RNA live-cell reporter based on an enzymatically inactive form of the small bacterial endonuclease Csy4, which binds to its cognate stem-loop with picomolar affinity. This system allows imaging of plant viral RNA genomes inside plasmodesmata and shows that potato virus X RNA remains accessible within the channels and is therefore not fully encapsidated during movement. We also combine Csy4-based RNA-imaging with interspecies movement complementation to show that an unrelated movement protein from tobacco mosaic virus can recruit potato virus X replication complexes adjacent to plasmodesmata. Therefore, recruitment of potato virus X replicase is mediated non-specifically, likely by indirect coupling of movement proteins and viral replicase *via* the viral RNA or co-compartmentalisation, potentially contributing to transport specificity. Lastly, we show that a 'self-tracking' virus can express the Csy4-based reporter during the progress of infection. However, expression of the RNA-binding protein *in cis* interferes with viral movement by an unidentified mechanism when cognate stem-loops are present in the viral RNA.

## Author Summary

The localisations of molecules within cells are intrinsically linked to their functions. In the past three decades major advances in cell biology have been enabled by the discovery

**Data availability statement:** The data that
support the findings of this study are publicly
available from BioImage Archive with the iden-
tifier https://doi.org/10.6019/S-BIAD1242.

**Funding:** Work in the JT lab is supported by the
Scottish Government Rural and Environment
Science and Analytical Services (RESAS)
division (https://www.gov.scot/publications/
environment-agriculture-and-food-strategic-re-
search-main-research-providers/). Parts of this
work were funded by the UK Biotechnology
and Biological Sciences Research Council
(https://www.ukri.org/councils/bbsrc/;grantBB/
M007200/1) and an award by the University
of St Andrews School of Biology Research
Committee (both to JT). MH was supported
by European Commission Erasmus+ funding
(https://erasmus-plus.ec.europa.eu/). ZB was
supported by a scholarship from the Eastbio
Doctoral Training Partnership funded by the
UK Biotechnology and Biological Sciences
Research Council (https://www.ukri.org/coun-
cils/bbsrc/;grantBB/T00875X/1). The funders
had no role in study design, data collection and
analysis, decision to publish, or preparation of
the manuscript.

**Competing interests:** The authors have
declared that no competing interests exist.

of fluorescent proteins and their use as genetically encoded 'tags' to label specific proteins
of interest in fluorescence microscopy. By contrast, it has been more difficult to fluores-
cently label other biomolecules in a similar specific manner, for instance ribonucleic acids
(RNAs). RNAs play important roles in gene expression and cellular regulation, as well as
constituting the genome of many medically or economically important viruses. Indirect
methods have been developed that use fluorescent proteins fused to sequence-specific
RNA-binding proteins to visualise localisations of particular RNAs in living cells. Here,
we characterise a system of this type that has a significantly increased sensitivity, and use it
to study the molecular mechanisms by which plant-infecting viruses spread through their
hosts. In addition to increasing our understanding of important crop pathogens, the sys-
tem should also be applicable to the study of other viruses in a variety of host organisms.

## Introduction

Plant viruses are major crop pathogens that contribute significantly to agroeconomic losses
and food insecurity [1, 2]. In order to spread through their hosts and ultimately, achieve
onward transmission, plant viruses transport their infectious genomes through intercellular
nano-channels called plasmodesmata (PD), which provide cytoplasmic continuity between
plant cells across the barrier of the cell wall [3–5]. This transport is mediated by virus-encoded
movement proteins (MPs). Plant virus movement systems are diverse and can consist of one
or multiple MPs. Nevertheless, some common features have emerged that are shared by many,
though not all MPs: 1) targeting to plasmodesmata; 2) increase of the plasmodesmal molecular
size exclusion limit, allowing passage of larger macromolecules, including the MPs them-
selves; 3) nucleic acid binding [3,6]. Studies of the localisations of MPs within host cells using
fluorescent proteins (FPs) have provided important insights into their functions and interac-
tions. By contrast, studying the localisation of viral nucleic acid cargoes – with RNA genomes
being more common among plant viruses - is more challenging [7, 8]. This has contributed to
significant knowledge gaps regarding fundamental aspects of plant virus movement such as
how replication and movement are linked [9] and the nature of transported ribonucleoprotein
(RNP) complexes.

For instance, none of the MPs characterised so far recognise their cognate viral genome
cargo sequence-specifically [3,6]. Movement often occurs early, within the first few hours of
infection when viral genomes are not necessarily the most abundant RNA in the cell. This
raises the question of how transport specificity is achieved [9–13]. Unlike MPs, viral RNA-
dependent RNA polymerases (replicases) and capsid proteins (CPs) often preferentially
interact with secondary structures in their cognate viral (v)RNA [14–18]. Therefore, transport
specificity might be achieved through movement being coupled to replication and/or encapsi-
dation [9]. Many viruses require their CP for cell-to-cell transport, and specific MP-CP inter-
actions required for movement have been described in such systems [3,6,9,19–21]. However,
other viruses move independently of CP [3,6], and many MPs are also able to non-specifically
transport unrelated viruses [22–24].

A variety of different systems have been developed for the sequence-specific live-cell
imaging of RNA molecules, which can be broadly classified into two categories. One group
of techniques requires cell permeation of all or some imaging components. This includes
directly labelled RNA [25], molecular beacons [26, 27] and other hybridisation-based probes
[28], fluorogenic compounds binding to RNA aptamers such as Spinach and Broccoli [29, 30],
and RNA-targeting, modified Cas9, which requires cell permeation of a synthetic DNA-RNA
hybrid PAMmer [31]. In intact plants, the low permeability of the cell wall generally makes use

of these types of imaging systems challenging, essentially limiting them to use in protoplasts or microinjection [7,8,25,32]. The other group of techniques uses only genetically encoded reporter components that can be expressed in walled plant cells, and principally comprises sequence-specific RNA binding proteins fused to FPs [7, 8]. Such genetically encoded RNA reporters either rely on the tagging of the RNA of interest with the cognate target sequence (typically stem-loops) [33–36], or the programming of suitable proteins to recognise a native RNA of choice [37–39]. The advantage of tagging RNAs with binding motifs is that the affinity of the reporter for the tag is usually well characterised and sensitivity can be modulated by varying the number of tags, typically between 6 – 24 but sometimes up to 96 to achieve single molecule sensitivity [40]. On the other hand, the tags can potentially influence RNA localisation and function [33,41–43]. In the case of vRNAs, tags might affect the regulation of viral replication and gene expression, which often rely on secondary structures and long-distance base-pairing [44–46], and thereby reduce infectivity. Programmable RNA binders on the other hand allow imaging of native RNAs at natural expression levels, but the sensitivity cannot be adjusted through changing the number of binding sites, and the affinity and specificity may be affected in unpredictable ways when protein engineering is required to modify sequence specificity [47]. In addition to single-stranded RNA (ssRNA) binding proteins, double-stranded RNA (dsRNA)-specific binders have also been used to localise viral replication intermediates [48, 49].

Previously, we have used the 'tunable' RNA binding domain from human Pumilio 1 protein coupled to bimolecular fluorescence complementation (PUM-BiFC) [37] for imaging the vRNA of potato virus X (PVX), the type member of the potexviridae and an important model system for triple gene block (TGB) MP systems, RNA silencing suppression and development of virus-based expression vectors and nanotechnology [50–52]. Likely due to the high RNA affinity of Pumilio ($k_D$ 0.5 nM for the wild type [47] - the affinity of the engineered variants was not experimentally determined), PVX vRNA could be successfully imaged with only two binding sites. This enabled the observations that PVX-induced inclusion bodies (historically termed 'X-bodies') accumulate vRNA and constitute viral replication compartments (VRCs), where the vRNA is arranged in circular 'whorls' around aggregates of the MP TGB1, and that small replication sites are also present at PD entrances, possibly for co-replicational delivery of progeny vRNA into the channels [38,53,54]. However, no vRNA was detected inside PD and this was interpreted as possibly being due to the low numbers of vRNA molecules transported, or their inaccessibility within movement RNPs. Other RNA imaging systems that have been used on plant viruses include the stem-loop binders MS2 capsid protein (MS2CP) on turnip crinkle virus with two tags [55] and a 22-amino acid peptide from bacteriophage $\lambda$ N protein ($\lambda N_{22}$) on potato virus A with 16 tags [56]. Neither of these studies observed labelling of vRNA inside PD. However, PD localisation was observed for a non-viral, nuclear-expressed messenger (m)RNA encoding the MP of tobacco mosaic virus (TMV) when tagged with 12 – 18 stem-loops for recruiting MS2CP, $\lambda N_{22}$ or the bacterial transcriptional antiterminator BglG [57, 58]. Similarly, MS2CP combined with 24 stem-loop tags also showed PD localisation of other non-viral mRNAs [59]. Thus, imaging vRNA at PD should be possible if it is accessible to RNA binding proteins.

Here, we characterise an RNA imaging system based on the bacterial endonuclease Csy4, which binds its cognate stem-loop with the highest currently known affinity for this kind of protein-RNA interaction (0.05 nM) [60]. Csy4 is a small (21.4 kDa) protein that plays a role in bacterial CRISPR (clustered regularly interspaced short palindromic repeats) immunity systems against bacteriophages. Csy4 processes pre-CRISPR transcripts into mature guide RNAs by recognising a specific stem-loop and cleaving the RNA immediately downstream of the dsRNA stem [61]. The endonuclease activity of Csy4 can be removed with a single

point mutation [H29A] (Csy4*) that does not affect the affinity for the stem-loop [61, 62]. This inactivated Csy4* has been used for RNA pull-down experiments to identify bound proteins, including in plants [62, 63]. We reasoned that its small size and high affinity might make Csy4* an advantageous reporter for vRNA imaging. We show that Csy4* for the first time enables visualisation of vRNAs inside PD, including for PVX and TMV tagged with just two stem-loops. We further use Csy4* to analyse vRNA localisation during interspecies viral movement complementation, including PD recruitment of PVX VRCs by TMV MP. Lastly, we describe a 'self-tracking' PVX genome expressing the Csy4* reporter during movement but also show that movement is inhibited when Csy4* is expressed *in cis* from a stem-loop-tagged virus, indicating that virus-expressed proteins may have preferential access to the vRNA.

## Results

### Fluorescent fusions of inactivated Csy4* are enriched at RNA-rich subcellular structures

In order to test the suitability of enzymatically inactive Csy4* as an RNA imaging system in plant cells, we initially generated four different GFP fusion constructs, with GFP fused either to the N- or the C-terminus of Csy4*, and both with, or without a nuclear localisation signal (NLS) for each orientation. In transient expression experiments in *Nicotiana benthamiana* leaves, both GFP-Csy4*-NLS and Csy4*-NLS-GFP fusions showed the expected nuclear localisation (S1A and S1B Fig). However, non-targeted GFP-Csy4* formed large aggregates in the cytoplasm (S1C Fig). Therefore, N-terminal fusions were not used for any further experiments. By contrast, non-targeted Csy4*-GFP showed nuclear and cytoplasmic localisation typical of soluble proteins, but was also observed enriched in the nucleolus and in small mobile granules within the cytoplasm (Fig 1A). Similar nucleolar enrichment was also observed for the nuclear-targeted fusions (S1A and S1B Fig).

To test if the Csy4*-GFP labelled granules corresponded to previously described cellular RNA granules, we performed co-localisation experiments with known fluorescent marker proteins. Only 18 ± 11% of Csy4*-GFP granules showed co-localisation with the processing

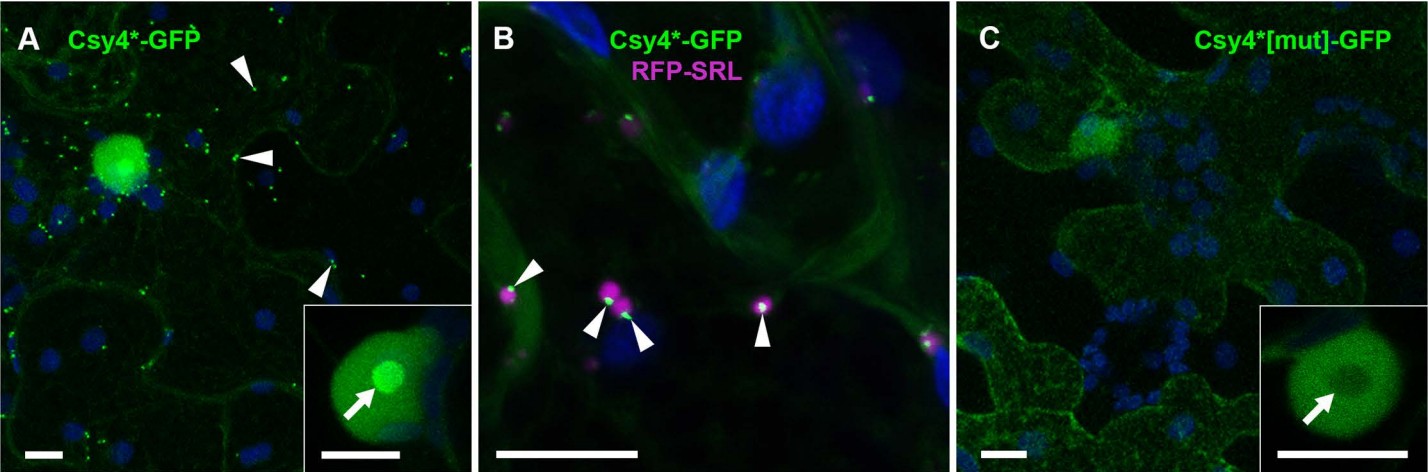

**Fig 1. Localisation of Csy4*-GFP fusions in the absence of tagged target RNAs.** (A) C-terminal GFP fusion without a nuclear localisation signal (Csy4*-GFP) shows nuclear and cytoplasmic localisation, including cytoplasmic motile granules (arrow heads). Inset shows enrichment in nucleolus (arrow). (B) Csy4*-GFP motile granules co-localise with periphery of peroxisomes labelled with RFP-SRL (Ser-Arg-Leu) (arrow heads). (C) C-terminal GFP-fusion of Csy4*[mut] (R114/115/118/119A) shows nuclear and cytoplasmic localisation with no motile granules. Inset shows depletion from nucleolus (arrow). All images are maximum intensity z-projections. GFP fluorescence shown green, RFP fluorescence shown magenta and chlorophyll auto-fluorescence shown blue. Scale bars, 10 μm.

(P-)body marker DECAPPING 1 (DCP1)-RFP [64], and only 25 ± 16% with the trans-acting silencing (tasi)RNA processing body marker RNA-DEPENDENT RNA POLYMERASE 6 (RDR6)-RFP [65] (S2 Fig). The stress granule marker RFP-RNA-BINDING PROTEIN 47 (RBP47) forms granules under heat stress conditions but is nucleo-cytoplasmically distributed under non-stressed conditions [64] and did not co-localise with the Csy4*-GFP granules (S2 Fig). We therefore tested co-localisation with markers of other small cellular bodies and found that Csy4*-GFP granules showed strong co-localisation (95 ± 5%) with the peroxisomal marker RFP-SRL [66] (Fig 1B). However, the Csy4*-GFP signal appeared as a smaller granule at the periphery of peroxisomes, rather than showing full co-localisation. Currently, the nature of these peroxisome-associated structures remains unclear.

To test if Csy4* localisations were related to its ability to bind RNA, we generated a mutated version, in which Arg residues 114, 115, 118 and 119 were all replaced with Ala (Csy4*[mut]). The strength of Csy4 RNA binding is largely due to the interaction of an Arg-rich helix with the major groove of the cognate stem-loop, and mutation of Arg115 and 119 within this helix reduces interaction with RNA 15,000-fold [60]. When we analysed localisation of Csy4*[mut]-GFP we observed nucleo- and cytoplasmic GFP distribution but notably a nucleolar depletion and absence of granule labelling (Fig 1C). The nucleolus is an RNA-rich environment [67], and nucleolar enrichment has also been observed for other RNA binding proteins used for RNA imaging, for instance MS2CP and $\lambda N_{22}$ [68]. The absence of Csy4*[mut]-GFP from the nucleolus and granules therefore indicates that the contrasting enrichment of Csy4*-GFP in these structures may be due to non-specific RNA binding, and the peroxisome-associated Csy4*-GFP granules may thus also be RNA-rich structures.

## Fluorescent Csy4* fusions label viral RNA at replication sites

To test the suitability of Csy4* for imaging viral RNAs *in vivo*, we expressed Csy4*-GFP or Csy4*-NLS-GFP in tissue infected with PVX tagged with four *csy* stem-loops at the same genomic position upstream of the CP where tags had previously been inserted for PUM-BiFC imaging [38], and expressing a red fluorescent protein (RFP) fusion of the CP (PVX.4x*csy*.RFP-2A-CP). With both constructs, GFP was localised to the 'X-body' in infected cells (Fig 2A and 2B), a large perinuclear structure generated within about one day after infection, which contains viral proteins and RNA and constitutes a VRC where replication and packaging of new virions occur [53]. Within the PVX VRC, Csy4*-GFP signal was distributed non-homogenously, with 'whorls' of fluorescence surrounding darker, non-labelled areas. This closely resembles PVX vRNA localisation previously observed with a PUM-BiFC reporter system, where the vRNA 'whorls' were found to surround aggregates of the viral movement protein TGB1 [53].

We next tested PVX constructs with no (PVX.RFP-2A-CP) or only two *csy* stem loops (PVX.2x*csy*.RFP-2A-CP) with the Csy4*-GFP reporter. With PVX.2x*csy*.RFP-2A-CP, the Csy4*-GFP localisation within the VRC was similar to the vRNA with 4x*csy* tags (Fig 2C). Thus, two cognate stem-loops are sufficient for vRNA imaging with Csy4*-GFP. The PVX VRC contains cytoplasm as well as multiple host organelles [53], therefore, we expected to see some Csy4*-GFP fluorescence within this structure also in the absence of any tags in the vRNA. This was indeed the case and the fluorescence was mostly diffuse as expected (Fig 2D). However, even in the absence of cognate stem-loops in the viral genome, the Csy4*-GFP reporter still weakly showed some of the characteristic localisation to 'whorls'. As vRNA is highly abundant, this might be due to some level of non-specific RNA association, similar to Csy4*-GFP localisation to the nucleolus. In agreement with this, when we instead used the non-binding Csy4*[mut]-GFP, only diffuse GFP fluorescence with no 'whorls' was observed in the VRC of PVX.2x*csy*.RFP-2A-CP (S3A and S3B Fig).

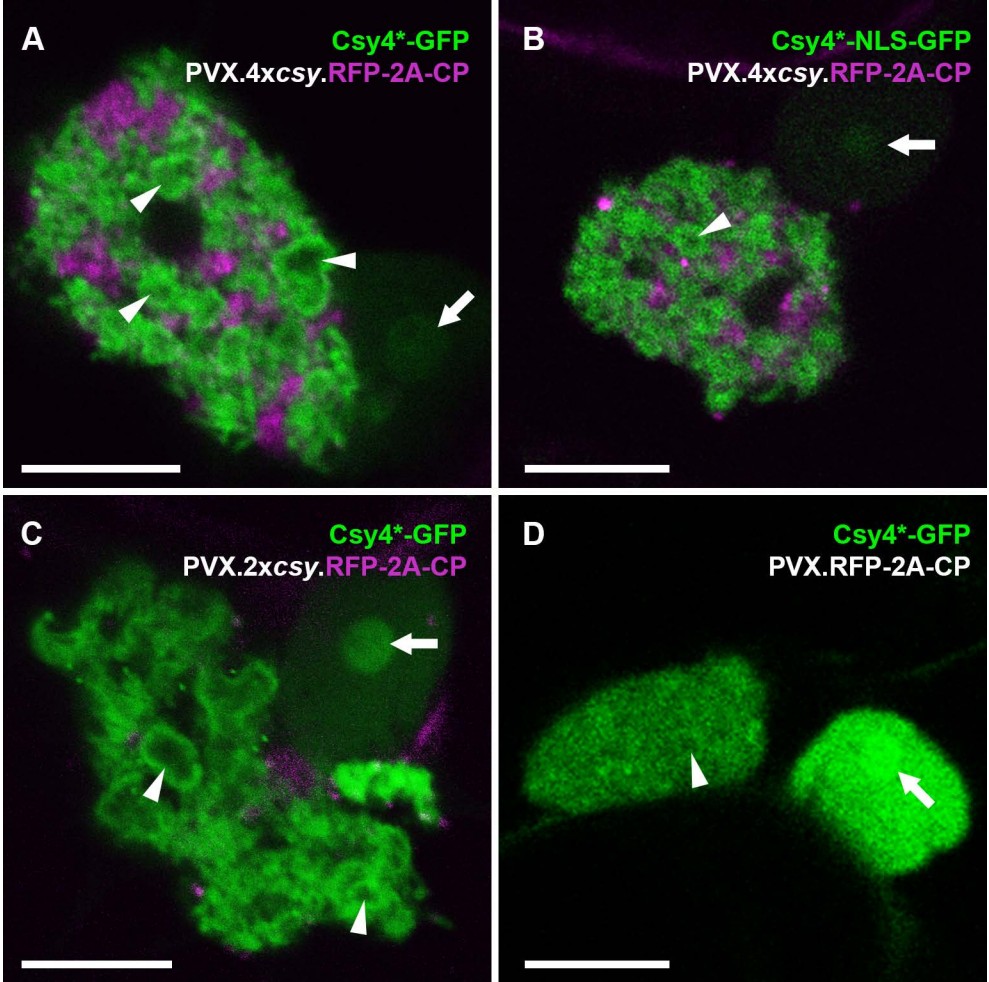

**Fig 2. Fluorescent Csy4\* fusions label PVX replication sites.** (A) Un-targeted Csy4\*-GFP decorates vRNA in the perinuclear 'X-body' of PVX tagged with four cognate *csy* stem-loops. Some of the vRNA is arranged in circular 'whorls' with dark centres (arrow heads), as previously described [53]. Faint GFP fluorescence is visible in the nucleoplasm, enriched in the nucleolus (arrow). (B) Nuclear-targeted Csy4\*-NLS-GFP similarly labels vRNA with 'whorls' in the 'X-body' of a four stem-loop tagged PVX. (C) Csy4\*-GFP shows a similar 'X-body' localisation when the PVX genome is tagged with only two *csy* stem-loops. (D) When the PVX vRNA is untagged, Csy4\*-GFP localisation within the 'X-body' is more diffuse, though very weak association with 'whorls' can sometimes still be identified (arrow head). By contrast, the nuclear and nucleolar (arrow) signal is comparatively stronger. (RFP channel not shown to more clearly show faint 'whorls'). GFP fluorescence shown green, RFP fluorescence shown magenta. All images are single z-sections. Scale bars, 10 μm.

## Csy4\* labels viral RNA inside plasmodesmata

When we examined the cell boundaries of cells expressing Csy4\*-GFP and infected with PVX.2x*csy*.RFP-2A-CP or PVX.4x*csy*.RFP-2A-CP we frequently observed GFP signal in static punctate structures at the cell wall (Fig 3A and 3B). At high magnification, these could be seen to be located within the otherwise non-fluorescent cell wall, between the two neighbouring layers of cortical cytoplasm (Fig 3C and 3D). These punctate structures co-localised with RFP-2A-CP, a known PD marker [69, 70]. No PD localisation of Csy4\*-GFP was observed in the absence of virus infection. When untagged PVX.RFP-2A-CP was used for infection, PD labelling was rarely observed and then, Csy4\*-GFP enrichment in PD compared with the adjacent cytoplasm

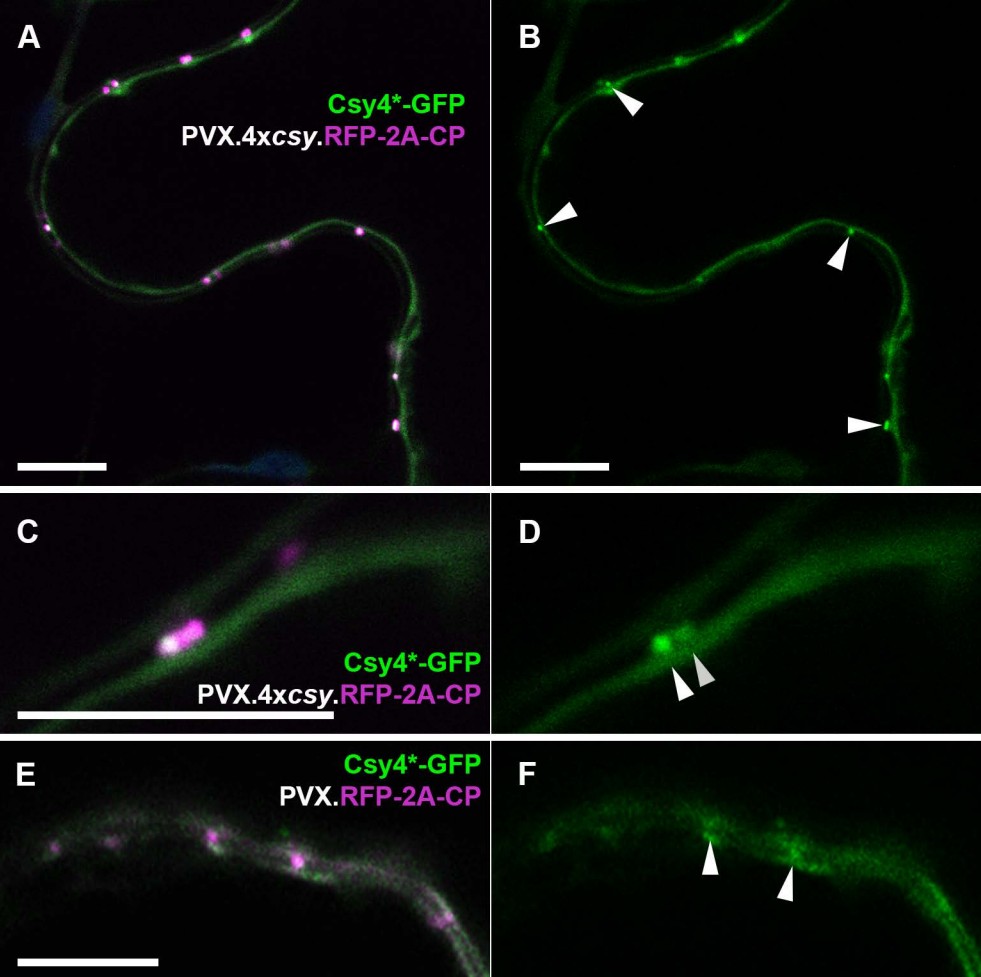

**Fig 3. A Csy4\*-GFP fusion labels PVX vRNA inside plasmodesmata.** (A, B) Cell wall boundary between two epidermal cells infected with PVX.4x*csy*.RFP-2A-CP and expressing Csy4\*-GFP. (A) RFP and GFP channel. RFP-2A-CP expressed from the virus labels plasmodesmata. (B) GFP channel only. Csy4\*-GFP co-localises with red plasmodesmata and can be seen extending into the inter-cellular wall space (arrow heads). (C, D) Higher magnification image of a cell boundary showing red CP fusion in the dark cell wall space between peripheral cytoplasms of neighbouring cells. (C) RFP and GFP channels. (D) GFP channel only. Csy4\*-GFP can clearly be seen to be concentrated in the inter-cellular cell wall space, i.e. the inside of plasmodesmata (arrow heads). (E, F) With an untagged PVX.RFP-2A-CP, co-localisation of Csy4\*-GFP with CP in plasmodesmata was rarely observed, and GFP enrichment in PD compared to adjacent cytoplasm was weaker than for a tagged virus (F, arrow heads). GFP fluorescence shown green, RFP fluorescence shown magenta. All images are single z-sections. Scale bars, 10 μm.

was weaker than for a tagged virus (Fig 3E and 3F). The non-binding Csy4\*[mut]-GFP did not detect RNA of tagged PVX.2x*csy*.RFP-2A-CP inside PD (S3C and S3D Fig).

## Csy4\* is more sensitive than other genetically encoded RNA imaging systems

In order to test the Csy4\*-GFP system on another virus and directly compare its sensitivity to other well-characterised RNA imaging systems such as PUM-BiFC [37, 38], MS2CP and its cognate RNA stem-loop (*ms2*) [33,55,57,68] and the λN$_{22}$ peptide binding an RNA stem-loop termed *boxB* [34,56,68], we produced a set of TMV constructs expressing RFP as an infection

marker, each tagged with only two binding sites for one of the respective imaging systems. The virus constructs lacked a CP, which should maximise access of the reporter constructs to vRNA, and tags were inserted just downstream of the RFP open reading frame (ORF), except for PUM-BiFC, where two different binding sites for engineered Pumilio homology domain fusions occur naturally within the replicase ORF [38]. For a more even comparison, all reporter constructs also lacked a nuclear targeting signal, which is commonly added to MS2CP and $\lambda N_{22}$ to reduce cytoplasmic background fluorescence [33,34,55,56,68]. With Csy4*-GFP, both localisation to the VRC and labelling of PD could be observed (Fig 4A and 4B). With PUM-BiFC, as reported before [38], preferential labelling of the VRC, with granular substructure, was easily observed. However, no labelling of plasmodesmata was detectable (Fig 4C and 4D). When MS2CP-GFP or $\lambda N_{22}$-GFP were expressed in tissue infected with TMV.ΔCP.RFP.2x*ms2* or TMV.ΔCP.RFP.2x*boxB*, respectively, no GFP localisation to PD was observed, and GFP fluorescence within VRCs was diffuse (Fig 4E-4H). Thus, two binding sites were sufficient for VRC imaging only with PUM-BiFC and Csy4*-GFP, and only the latter could detect vRNA in plasmodesmata, indicating that it is the most sensitive amongst these RNA imaging systems.

## Multiple *csy* tags have only a moderate negative effect on viral infectivity

The requirement for only a small number of tags may be advantageous as extensive addition of secondary structure elements could affect RNA localisation and function, particularly in the case of vRNAs where infectivity might be reduced. When generating tagged virus constructs, we observed that multiple insertions of *csy* tags predominantly occurred in tandem, i.e. either all in forward or all in inverted orientation. Although the reason for this is unclear, we decided to use constructs with multiple tags to test the effect on infectivity directly by comparing PVX.2x*csy*.RFP-2A-CP and PVX.12x*csy*.RFP-2A-CP. Both viral replicons produced local and systemic fluorescent infection sites. However, despite the absence of Csy4* protein binding to the stem-loops, local lesions produced by the virus with 2x tags were larger at 6 days post inoculation (dpi) than those of the 12x tagged virus (Fig 5A-5C). Furthermore, systemic infection by the 12x-tagged virus was reduced compared to the 2x-tagged virus, showing significantly reduced RFP fluorescence (Fig 5D and 5F). At 14 dpi, reverse transcription polymerase chain reaction (RT-PCR) detected full-length *csy*.RFP-2A-CP cassettes in systemic leaves infected with 2x-tagged virus (17/18 plants = 94%, n = 3 biologically independent experiments) (Fig 5G). For the 12x-tagged virus, all systemic infections (18/18 plants = 100%, n = 3) produced *csy*.RFP-2A-CP PCR products that were larger than for the 2x-tagged virus, though some products were either slightly smaller than the 12x*csy* positive control, or produced a double band (4/18 plants = 22%). This suggests that not all systemic infections maintained the full 12 stem-loops, but none completely lost the tags either. Thus, 12x*csy* stem-loops are maintained relatively stably through systemic infection and increasing the number of hairpin tags was only mildly detrimental to viral infectivity.

## TMV 30k movement protein non-specifically recruits PVX replication complexes to plasmodesmata

PVX and TMV, as well as other plant viruses target replication complexes to PD, which may increase the speed or specificity of cell-to-cell movement during early infection stages [54,71,72]. This raises the question of how the viral replicase is recruited, with the only known PD-targeting activity mediated by the movement proteins [9]. In PVX, interaction between TGB2 movement protein and the replicase is required for VRC targeting [73]. However, movement complementation by heterologous movement proteins has been frequently

reported for various plant viruses [23], suggesting that in such artificial systems either replication complexes are not targeted to PD, or if they are, the replicase is likely to be recruited by a mechanism other than specific interaction between cognate MP(s) and replication enzymes. Since RNA localisation had so far not been experimentally observed during cross-species movement complementation, we decided to use Csy4*-GFP to analyse movement complementation of PVX by TMV 30k movement protein. TMV 30k was chosen because both viruses target replication complexes to PD, and also because it is the simplest known movement protein system, requiring no other viral proteins for local cell-to-cell transport [74], and has been shown to complement movement of various unrelated viruses, including PVX, and even non-plant viruses [22,23,75].

We replaced the complete triple gene block within a 4x*csy*-tagged PVX construct with TMV 30k such that the start and stop codons of the 30k ORF exactly replaced the TGB1 start and TGB3 stop codons, respectively (PVX.ΔTGB.30k.4x*csy*.RFP-2A-CP). This virus was capable of cell-to-cell movement (Fig 6A), similar to previously reported movement complementation of PVX TGB mutants by tobamovirus 30k proteins [75–78] though spread was slower than PVX with a native triple gene block (not quantified). No systemic movement was observed.

The PVX.ΔTGB.30k.4x*csy*.RFP-2A-CP infections produced perinuclear VRCs which contained both Csy4*-GFP vRNA and 165k-GFP replicase reporters (Fig 6B and 6C), similar to native PVX. However, the vRNA was not arranged in circular 'whorls', as expected for PVX lacking TGB proteins [38,53,69]. At the periphery of infection sites, the Csy4*-GFP RNA reporter could be observed enriched inside PD, but with no concurrent enrichment of RFP-CP above cytoplasmic levels (Fig 6D), unlike native PVX (Fig 3) but as expected for movement mediated by TMV 30k. Weak RFP signal inside PD may have been due to virus-induced gating or interaction of RFP-2A-CP with PVX RNA, but lack of enrichment compared with the Csy4*-GFP signal indicates a low CP/RNA ratio and therefore likely no significant encapsidation. vRNA was also observed in peripheral structures adjacent to PD ahead of the infection front (Figs 6E, S4A and S4B). The 165k-GFP replicase reporter was also recruited to the peripheral structures adjacent to PD (Figs 6F, S4C and S4D). Thus, PVX replication complexes are recruited to PD even when PD targeting and movement are mediated by the heterologous TMV 30k protein, indicating that the recruitment is not virus-specific. To test if TMV 30k might mediate PVX VRC recruitment by unexpectedly interacting directly with PVX replicase or CP, the respective proteins were co-expressed by agroinfiltration in the absence of virus infection. In the presence of 30k, PVX replicase remained confined to the nucleus and CP nucleo-cytoplasmic, with neither being enriched at PD (S5 Fig). Thus, 30k was not able to recruit either PVX protein directly.

## A 'self-tracking' PVX construct demonstrates preferential access to vRNA when Csy4*-GFP is expressed *in cis*

As the Csy4*-GFP fusion is relatively small (1335 bp, 50.2 kDa) we engineered a 'self-tracking' virus that expresses Csy4*-GFP *in cis*, highlighting its own RNA during virus spread, to avoid the necessity for co-expression of Csy4*-GFP. For this, we replaced the RFP ORF in a PVX.RFP-2A-CP construct with Csy4*-GFP. In PVX 'overcoat' constructs translationally fusing foreign ORFs to the 5' end of the CP ORF, the 2A peptide from foot and mouth disease virus (FMDV) is used as a linker [69]. 2A mediates a recoding event termed 'stop-carry-on-translation' in which ribosomes sometimes release the nascent polypeptide, up to and including the penultimate 2A residue, without terminating translation [79], resulting in a mixed population of free and fused foreign protein and PVX CP. The FMDV 2A peptide mediates a ratio of < 50% co-translational separation in 'overcoat' PVX [69]. To

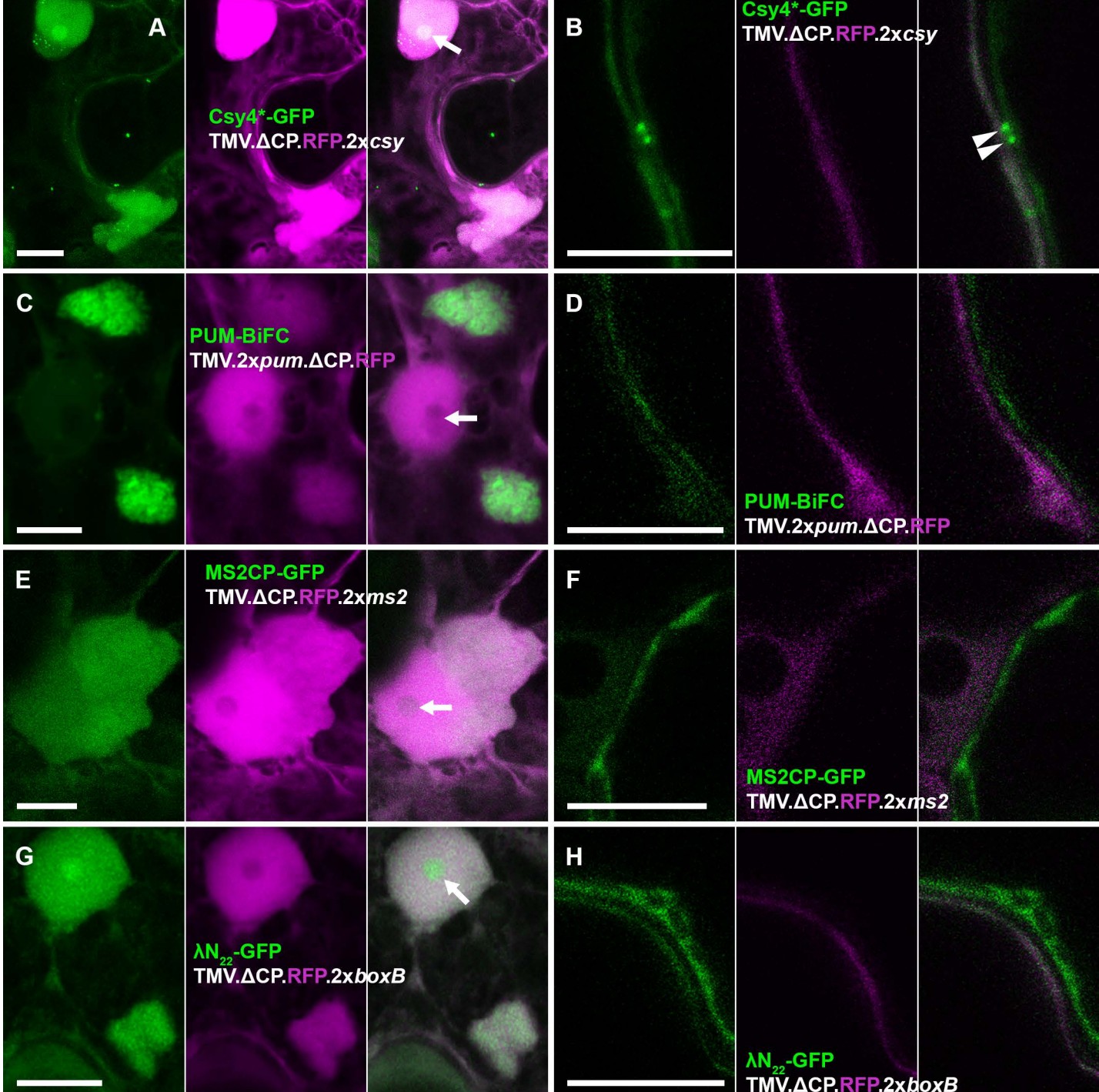

**Fig 4. Only Csy4*-GFP detects TMV vRNA containing two binding sites inside plasmodesmata.** (A, B) Csy4*-GFP system. (C, D) PUM-BiFC system. (E, F) MS2CP-GFP system. (G, H) $\lambda N_{22}$-GFP system. (A, C, E, G) Representative images of TMV VRCs near the nucleus. Note that Csy4*-GFP and $\lambda N_{22}$-GFP are enriched in the nucleolus, whilst PUM-BiFC is depleted and MS2CP neither enriched nor depleted compared with the nucleoplasm (arrows). (B, D, F, H) Representative images of cell boundaries at the leading edge of viral lesions. Only Csy4*-GFP shows enrichment in punctate structures within the non-fluorescent cell wall layer between adjacent cells, corresponding to plasmodesmata. Each image shows the GFP channel on the left, RFP channel in the middle and merged GFP and RFP channels on the right. GFP fluorescence shown green, RFP fluorescence shown magenta. Panels (A, C, E, G) are maximum intensity z-projections, panels (B, D, F, H) are individual z-sections. Scale bars, 10 μm.

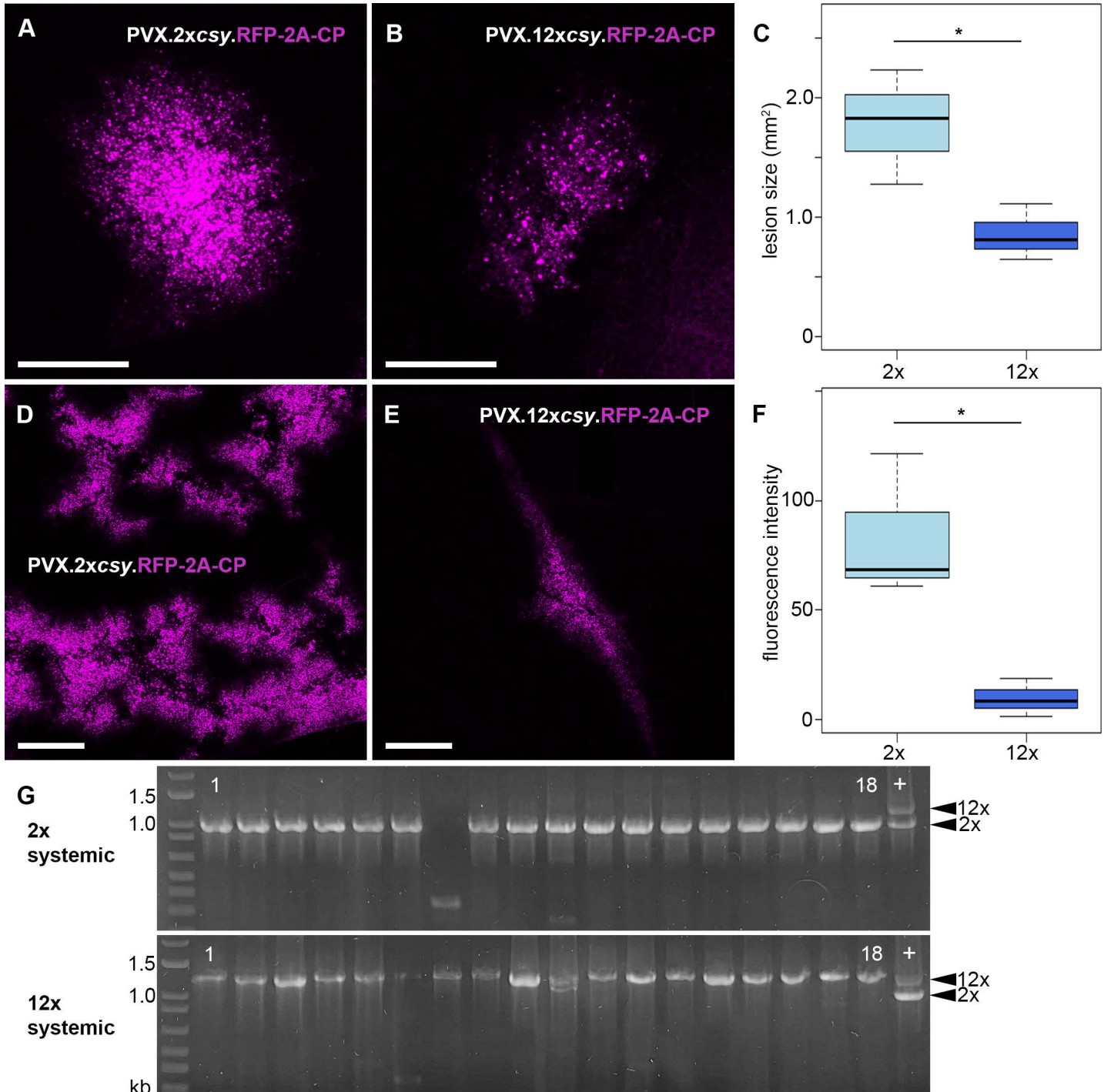

**Fig 5. 12xcsy stem-loop tags cause a moderate reduction in PVX infectivity.** (A) Representative (i.e., close to mean lesion size) infection site of PVX.2x*csy*.RFP-2A-CP on inoculated leaf, 6 days post inoculation (dpi). (B) Representative (i.e., close to mean lesion size) infection site of PVX.12x*csy*.RFP-2A-CP on inoculated leaf, 6 dpi. (C) Box plot of lesion sizes on inoculated leaves, 6 dpi. 2x: PVX.2x*csy*.RFP-2A-CP (143 infection sites), 12x: PVX.12x*csy*.RFP-2A-CP (85 infection sites). Bold line represents median, box 25% quartile, bars standard deviation (SD) (t-test: n = 3, p = 0.04044, * = significant). (D) Representative (i.e., close to mean fluorescence intensity) systemic infection site of PVX.2x*csy*.RFP-2A-CP, 14 dpi. (E) Representative (i.e., close to mean fluorescence intensity) systemic infection site of PVX.12x*csy*.RFP-2A-CP, 14 dpi. (F) Box plot of fluorescence intensities on systemically infected leaves, 14 dpi. 2x: PVX.2x*csy*.RFP-2A-CP, 12x: PVX.12x*csy*.RFP-2A-CP (36 leaves each). Bold line represents median, box 25% quartile, bars SD (t-test: n = 3, p = 0.01957, * = significant). RFP fluorescence shown magenta. All images are maximum intensity z-projections. Scale bars, 1 mm. (G) RT-PCR analysis of systemically infected leaves at 14 dpi. A segment of the PVX genome from the end of the TGB3 ORF to the beginning of the CP ORF was amplified. Arrow heads on right indicate expected product sizes for 2x and 12x tagged viruses, respectively. 1 to 18: three biological replicates with six plants each. +: positive control, plasmids used for inoculation were amplified with the same primers and a mixture of products from 2x and 12x tagged constructs loaded as size standard.

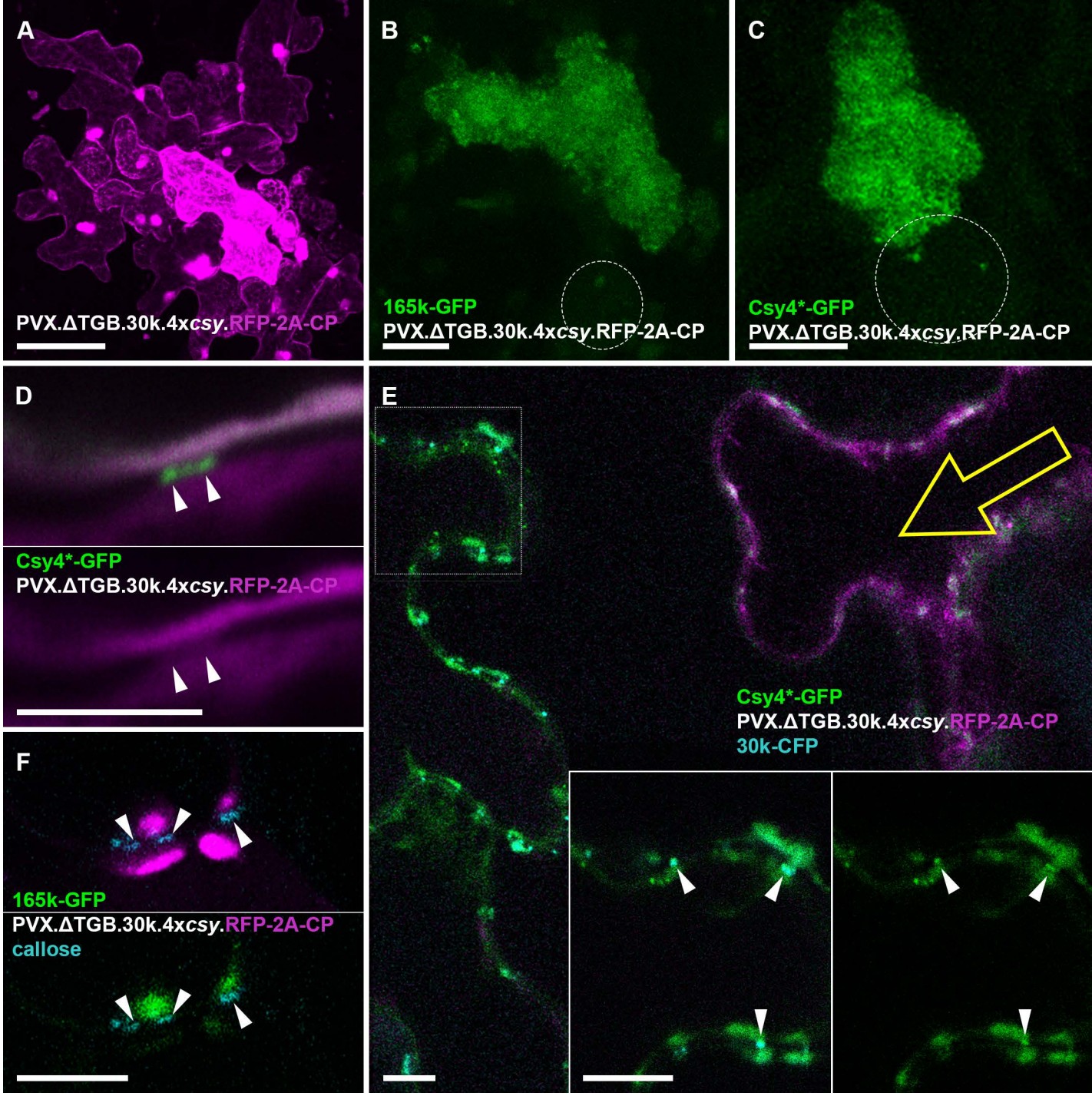

**Fig 6. PVX VRCs are recruited adjacent to plasmodesmata when intercellular movement is complemented by TMV 30k protein.** (A) Fluorescent infection site of PVX.ΔTGB.30k.4x*csy*.RFP-2A-CP. (B) Localisation of a PVX replicase (165k)-GFP fusion to the VRC in a PVX.ΔTGB.30k.4x*csy*.RFP-2A-CP infected cell. Circle indicates location of nucleus. (C) Localisation of vRNA labeled by Csy4*-GFP within the VRC in a PVX.ΔTGB.30k.4x*csy*.RFP-2A-CP infected cell. No circular arrangement of vRNA in 'whorls' observed. Circle indicates location of nucleus. (D) Csy4*-GFP labels vRNA of PVX.ΔTGB.30k.4x*csy*.RFP-2A-CP inside plasmodesmata (arrow heads), but unlike during native PVX transport, RFP-2A-CP is not co-enriched within plasmodesmata. (E) Csy4*-GFP vRNA imaging at the leading edge of a PVX.ΔTGB.30k.4x*csy*.RFP-2A-CP lesion. Yellow arrow indicates approximate direction of viral spread. The inset is a separate higher magnification image acquired at the boxed region. vRNA can be observed in 'caps' at plasmodesmata entrances, as well as inside the channels labelled with 30k-CFP fusion expressed *in trans* (arrow heads). (A full maximum projection of the same image is shown in S4A and S4B Fig to confirm the presence of virus-expressed RFP in the cell on the left.) (F) 165k-GFP replicase marker located in 'caps' at the entrances of plasmodesmata labelled with aniline blue-staining of callose. (The full image from which panel F is derived

is presented in S4C and S4D Fig and shows multiple similar 'caps' containing 165k and CP.) GFP fluorescence shown green, RFP fluorescence shown magenta, CFP and aniline blue fluorescence shown cyan. Panels (A-C) are maximum intensity z-projections, (E) is a projection of two z-sections and (D, E inset and F) are single z-sections. Scar bars, 100 μm (A), 10 μm (B-C, E-F), 5 μm (D).

ensure expression of predominantly unfused Csy4*-GFP and CP, we replaced the FMDV 2A with a related 2A sequence from thosea asigna virus (T2A), which mediates 99% separation [80]. The resulting viral replicon, PVX.Csy4*-GFP-T2A-CP was systemically infectious (S6 Fig), strongly expressing Csy4*-GFP (Fig 7). RT-PCR with PVX-specific primers confirmed that a 1620 bp product corresponding to a complete Csy4*-GFP-T2A-CP cassette was present in systemically infected leaves from most inoculated plants (15/18 plants = 83%, n = 3 independent biological replicates) (S6 Fig). GFP-specific Western blots from systemic leaves showed bands at approximately 52, 29 and 27 KDa, corresponding to Csy4*-GFP-T2A (52.2 kDa), GFP-T2A (29.1 kDa) and GFP (27 kDa), respectively (Fig 7A). The latter two could be due to proteolytic cleavages (e.g. at the linker peptide connecting Csy4* and GFP, or at T2A), or partial loss of the Csy4*-GFP-T2A insert in a sub-population of viral genomes. Importantly, no 77.3 kDa band corresponding to the full-length Csy4*-GFP-T2A-CP fusion was observed, indicating near-complete co-translational separation into Csy4*-GFP-T2A + CP at the T2A linker.

In PVX.Csy4*-GFP-T2A-CP lesions, GFP fluorescence was enriched in nucleoli (Fig 7C), similar to when Csy4*-GFP was expressed by agroinfiltration. GFP fluorescence was also enriched in VRCs and plasmodesmata (Fig 7C-7E) and the mobile granules were also observed (Fig 7E). Thus, localisations of Csy4*-GFP expressed *in cis* corresponded to those after expression *in trans*. At the leading edge of the spreading infection site, vRNA labelling could be observed in plasmodesmata projecting out into the cell wall from the most recently infected cells (Fig 7E). The association of the reporter with vRNA inside PD even in the absence of hairpin tags was notably stronger than when Csy4*-GFP was co-expressed *in trans* with an untagged PVX.RFP-2A-CP, where even weak PD localisation was extremely rare (Figs 3E, 3F and 7D). The strong recruitment of Csy4*-GFP by PVX RNA even without *csy* tags facilitated co-localisations with transiently expressed PVX proteins (S7 Fig). TGB1-RFP, RFP-TGB2 and RFP-CP were all co-localised with vRNA inside PD, whilst TGB3-RFP localised to PD-adjacent membrane structures that represent localised replication sites, as previously observed [54]. Within VRCs, RFP-TGB2 and RFP-CP surrounded vRNA 'whorls' whilst 165k replicase localised to granular structures amongst the vRNA, again confirming previous observations [53].

Unexpectedly, when we attempted to complete the 'self-tracking' PVX construct by inserting *csy* tags at the same position as in all previous PVX constructs, directly upstream of the ORF encoding the translational fusion to CP, just two *csy* hairpins (PVX.2x*csy*.Csy4*-GFP-T2A-CP) resulted in a severe inhibition of viral cell-to-cell spread. Fluorescent infection sites were observed and showed VRCs with typical vRNA 'whorls', as well as nucleolar localisation of GFP fluorescence (Fig 7G). However, there was a strong reduction in the number of infection sites (0.3 ± 0.2 lesions/μg DNA for PVX.2x*csy*.Csy4*-GFP-T2A-CP, compared with 1.7 ± 0.2 lesions/μg DNA for PVX.Csy4*-GFP-T2A-CP, n = 3 independent experiments). Lesions of PVX.2x*csy*.Csy4*-GFP-T2A-CP also remained very small, typically confined to single cells (68% of infection sites) or occasionally small clusters, at 6 dpi (Fig 7F and 7M). In small infected cell clusters, PD labelling could be observed at cell boundaries (Fig 7H). Systemic leaves showed no GFP fluorescence at 14 dpi (S6 Fig) and RT-PCR either did not detect PVX at all (5/17 plants = 29%) or produced a ~500 bp band indicative of a loss of most of the Csy4*-GFP insert (12/17 plants = 71%, n = 3 independent biological experiments) (S6 Fig).

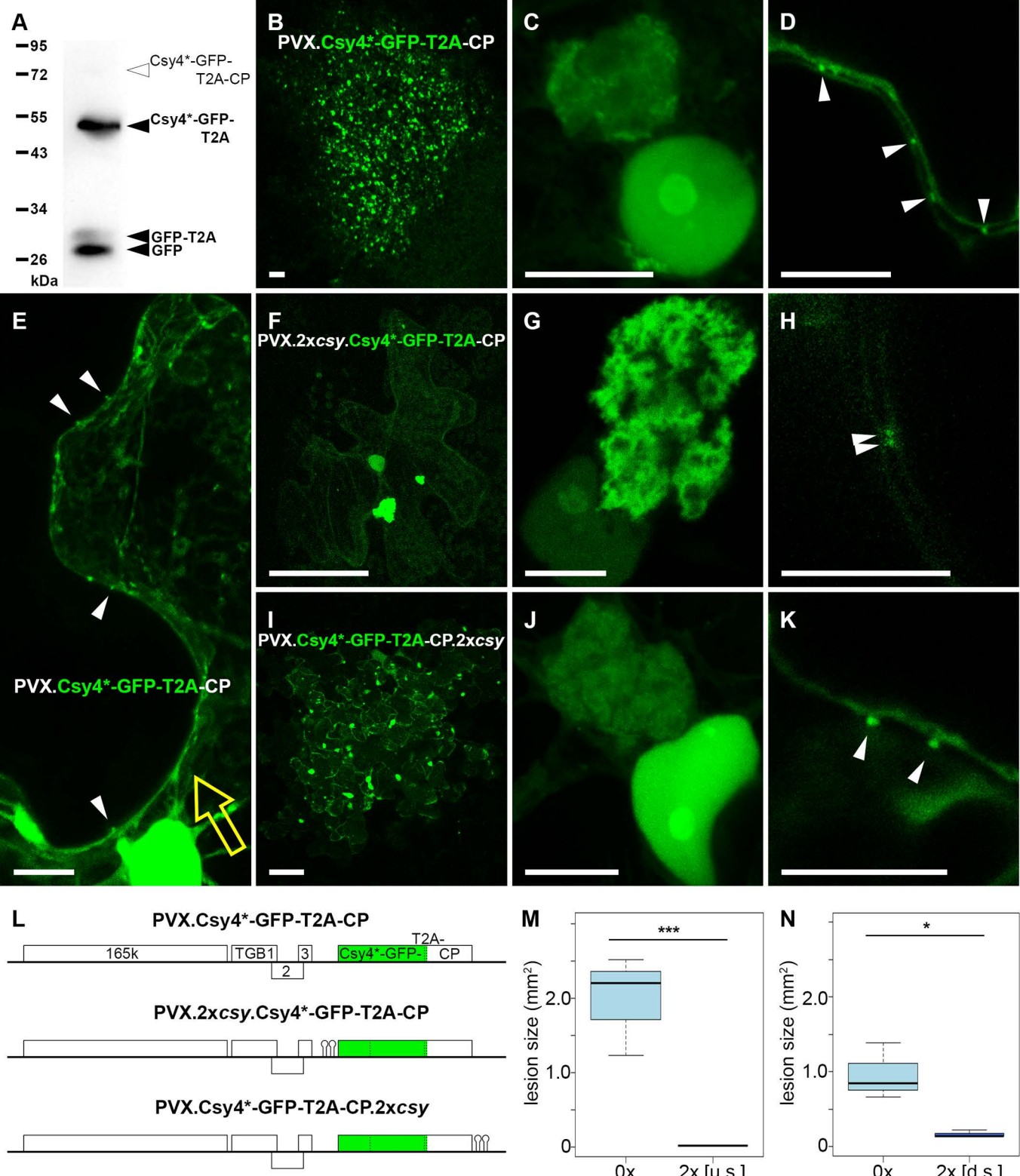

**Fig 7. 'Self-tracking' PVX constructs demonstrate preferential access of Csy4\*-GFP expressed from the viral genome *in cis*.** (A) Representative Western blot of total protein from leaf systemically infected with untagged PVX.Csy4\*-GFP-T2A-CP and probed with antibody against GFP. Black arrow heads indicate expected sizes of free GFP, GFP-T2A and Csy4\*-GFP-T2A fusion proteins. White arrow head indicates expected size of Csy4\*-GFP-T2A-CP fusion. (B-E) Untagged PVX.Csy4\*-GFP-T2A-CP. (B) Representative infection site; (C) VRC; (D) vRNA inside plasmodesmata (arrow heads). (E) Leading edge of

an infection site, yellow arrow indicates approximate direction of viral spread. vRNA can be observed in plasmodesmata projecting from the most recently infected cell (arrow heads). (F-H) PVX.2x*csy*.Csy4*-GFP-T2A-CP tagged directly upstream of the Csy4* ORF. (F) Representative single-celled infection site; (G) VRC; (H) vRNA inside plasmodesmata (arrow heads). (I-K) PVX.Csy4*-GFP-T2A-CP.2x*csy* tagged immediately downstream of the CP ORF. (I) Representative small multi-cellular infection site; (J) VRC; (K) vRNA inside plasmodesmata (arrow heads). (L) Schematic representation of 'self-tracking' PVX constructs (not to scale). (M) Comparison of lesion sizes of untagged PVX.Csy4*-GFP-T2A-CP (0x; 118 infection sites) and upstream-tagged PVX. 2x*csy*.Csy4*-GFP-T2A-CP (2x [u.s.]; 25 infection sites). Bold line represents median, box 25% quartile, bars SD (t-test: n = 3, p = 0.0009156, *** = significant). (N) Comparison of lesion sizes of untagged PVX.Csy4*-GFP-T2A-CP (0x; 134 infection sites) and downstream-tagged PVX.Csy4*-GFP-T2A-CP.2x*csy* (2x [d.s.]; 24 infection sites). (t-test: n = 3, p = 0.02117, * = significant). Experiments shown in panels M and N were conducted separately with identical conditions (inoculating first two true leaves of four week-old plants, leaves imaged at 6 dpi), but at different times of the year, and are therefore presented and analysed separately. Note that the upstream-tagged virus produced mostly (17/25 = 68%) single-celled infection sites when the untagged control had larger lesions (F, M), whilst the downstream-tagged virus produced mostly (18/24 = 75%) small multicellular infection sites even though the control had smaller lesions at this time (I, N). All images are maximum intensity z-projections with the exception of (D, G, H, K) which are individual z-sections. Scale bars, 100 μm (B, F, I), 10 μm (C-E, G-H, J-K).

Since PVX tolerates both insertion of up to 12x*csy* hairpin tags at the same genomic position, or viral expression of Csy4*-GFP-T2A if the virus does not contain *csy* hairpins, we decided to also compare the effect of Csy4*-GFP expression *in trans* from agroinfiltration on PVX.RFP-2A-CP constructs without tags and with 2x*csy* tags comparable to PVX.2x*csy*.Csy4*-GFP-T2A-CP. As had previously been observed, infection sites of both untagged and tagged virus expanded into Csy4*-GFP-expressing tissue. When comparing lesion sizes of untagged and tagged virus in Csy4*-GFP-expressing tissue to free GFP-expressing tissue as a control, there was a reduction in lesion size for the tagged virus on Csy4*-GFP (S8 Fig). However, whilst 2-way ANOVA did find a significant interaction between the number of stem-loop tags and the expressed GFP construct, pairwise comparisons of PVX.2x*csy*.RFP-2A-CP on Csy4*-GFP to the other combinations were not statistically significant. Thus, whilst binding of *trans*-expressed Csy4*-GFP to *csy* stem-loops seems to have a negative impact on lesion size, the *cis*-expression of Csy4*-GFP-T2A from a tagged vRNA causes a dramatically stronger inhibiting effect.

Because the position of the *csy* stem-loop tags in the genome might be important for the effect on infectivity, we further modified the 'self-tracking' construct to insert a 2x*csy* tag immediately downstream of the Csy4*-GFP-T2A-CP ORF stop codon, a preferred tagging position for cellular mRNAs [42]. The resulting virus, PVX.Csy4*-GFP-T2A-CP.2x*csy*, still showed only a low infectivity, with highly variable lesion numbers (0.4 ± 1.8 lesions/μg DNA), but the majority (75%) of infection sites was multi-cellular (Fig 7I), though still significantly smaller than those of untagged PVX.Csy4*-GFP-T2A-CP (Fig 7N). The sub-cellular localisations of GFP fluorescence observed with this construct corresponded to those with other 'self-tracking' constructs (Fig 7J and 7K). No GFP was observed in systemic leaves with this construct (S6 Fig; not quantified).

## Discussion

The bacterial RNA endonuclease Csy4 has previously been used to process multiplexed guide RNAs for genome editing [81] and for controlling mRNA stability, including as a fluorescent reporter for micro-RNA expression [82, 83]. Its enzymatically inactive form Csy4* has been used to identify RNA-associated proteins by pull-down of tagged RNAs [62, 63]. Here, we demonstrate that Csy4* also works as a high-sensitivity, genetically encoded reporter for *in vivo* localisation of viral RNA. Similar to PUM-BiFC [37, 38], Csy4* affinity for its cognate RNA recognition sequence is sufficiently high that vRNAs can be efficiently imaged when tagged with just two recognition sequences. This contrasts with analogous RNA imaging systems based on the specific interaction of an RNA binding protein with its cognate stem-loop, such as MS2CP and $\lambda N_{22}$, where two tags were insufficient for efficient recruitment by vRNA [55]; this study.

However, Csy4* is able to also label vRNA inside PD. Previously, a nuclear expressed, non-viral mRNA encoding the TMV *30k* ORF was shown to co-localise with the 30k protein in mobile granules and at PD, using the MS2CP system and 12 stem-loops downstream of the *30k* stop codon [57]. In analogous experiments tagging a non-viral *30k* mRNA with either, 16 *boxB* stem-loops recognised by λN$_{22}$, or 18 stem-loops recognised by BglG, both systems also co-localised the mRNA with 30k protein in mobile granules and at PD. Additionally, the BglG system allowed demonstration of intercellular mRNA movement by co-transport of BglG-GFP in the presence of tagged, but not untagged *30k* mRNA [58]. However, none of the MS2CP, λN$_{22}$ or BglG systems have so far been successfully used to image virus-produced RNAs at PD.

The dissociation constant for Csy4 bound to the *csy* stem-loop is 0.05 nM, one of the highest affinity small protein-RNA interactions known [60]. This is about two orders of magnitude stronger than MS2CP (1 – 6.4 nM [84, 85]), λN$_{22}$ (22 nM [86]), BglG (1.4 – 3.2 µM [87–89]), and other stem-loop binding proteins that have been used for RNA imaging such as bacteriophage PP7 coat protein (1.6 nM [85,90]) and the human U1 small nuclear ribonucleoprotein A (10 nM [91]). The wild-type Pumilio RNA binding domain binds its target sequence with about ten-fold lower affinity (k$_D$ 0.5 nM) [47] than Csy4. However, for imaging PVX vRNA, Pumilio domain variants engineered to bind a target sequence derived from the TMV genome were used [38] and the affinity of these modified protein domains was not experimentally tested. Pumilio homology domains engineered for altered sequence specificity can have considerably and unpredictably varying affinities for their target sequences, with dissociation constants between 0.05 nM, similar to Csy4, and 18 nM, 360-fold lower than Csy4, but the majority of variants exhibited either unchanged or reduced affinities [47]. Thus, the ability of Csy4* to label vRNA inside PD may be due to its higher affinity for its cognate target sequence compared with the other imaging systems. Additionally, in the case of stem-loop tags which are all derived from non-plant organisms, their correct folding in a plant cell environment, particularly in tandem repeats, may also play a role in determining sensitivity.

On the other hand, PD are extremely small, with a diameter of ~50 nm and an estimated free cytoplasmic diameter of only ~3 – 4 nm, approximately the Stoke's radius of a 27 kDa GFP molecule [5,92–94]. Correspondingly, the molecular size exclusion limit of PD is between ~10 - 80 kDa, depending on developmental and physiological conditions [92,95,96], although presumably this must be further increased during transport of several kb-long vRNAs, which have molecular weights in the range of MDa. Nevertheless, the ability of RNA reporters to track vRNA inside PD may be related to their size. Compared to the PUM-BiFC complex consisting of two Pumilio homology domains and the fluorescent protein (~108 kDa total), Csy4*-GFP (50.2 kDa) is significantly smaller. On the other hand, MS2CP-GFP (~41 kDa), λN$_{22}$-GFP (~30 kDa) and BglG-GFP (~33.5 kDa) are even smaller. Thus, given that PVX tolerated 12x*csy* stem-loops and PD-localised mRNAs were imaged with 12 – 18 stem-loops with the MS2CP, λN$_{22}$ and BglG systems, it seems feasible that these should also be able to detect vRNA at PD when more stem-loops are used.

In PVX, where the CP is required for movement and co-localises within PD, the encapsidation signal is located near the 5' end of the genomic RNA [15,50,97] and this genomic region is also required for movement [98]. To our knowledge, there is so far no evidence for encapsidation of PVX subgenomic RNAs. Therefore, we assume that when Csy4*-GFP is localised to PD, this is in association with genomic PVX vRNA. By contrast, for TMV and the movement-complemented PVX.ΔTGB.30k.4x*csy*.RFP-2A-CP, where 30k does not require a CP for movement and has already been shown to recruit its own messenger RNA [57], PD-localised Csy4*-GFP might be bound to both, genomic or subgenomic vRNAs.

Unexpectedly, Csy4*-GFP also showed association with 'whorls' in PVX VRCs and with PD when the virus was not tagged with cognate stem-loops. This non-specific recruitment was both rare and weak when the reporter construct was expressed *in trans* by agroinfiltration, but became noticeably more common and stronger for Csy4*-GFP *cis*-expressed from the viral genome. As the high RNA affinity of Csy4 is mainly due to interactions with the backbone of the major groove in the cognate stem-loop [60], non-specific binding is likely due to interactions with other RNA secondary structures. This is in agreement both, with Csy4* enrichment in the nucleolus, which contains structured ribosomal and small nucleolar (sno)RNAs, and with the absence of nucleolar and PD localisations when a Csy4 variant was used that contained mutations in the arginine residues interacting with the RNA major groove (Figs 1C and S3). Affinity of Csy4 for non-cognate stem-loops is several orders of magnitude lower than for its target sequence (which is still in the nM to μM range) [60]. However, in a complex cellular environment, the localisation of the Csy4*-GFP reporter is likely the result of a steady-state between binding and dissociation events and even low-affinity non-specific binding may lead to an enrichment in cellular regions containing structured RNA. It could perhaps be expected that non-specific recruitment to PD would also occur in uninfected cells due to trafficking of endogenous RNAs, but we did not observe this. Possibly, endogenous RNAs are less abundant, only some traffic through PD [99] and they may not accumulate within PD to the same level as viruses transported by their movement proteins. Nevertheless, Csy4*-GFP should have strong potential for imaging mobile plant mRNAs.

The ability of Csy4* fusions to detect PVX and TMV vRNA inside plasmodesmata provides novel insights into the mobile RNP forms of these viruses. PVX requires its capsid protein (CP) for transport [100], which is co-transported cell-to-cell [70,101]. By contrast, TMV does not require its CP for local transport [74]. In agreement with these established findings, PVX CP was co-enriched with Csy4*-GFP inside PD in native PVX infections, but not when the TGB proteins were replaced by TMV 30k (Fig 6D).

There has been debate over the role of CP in PVX movement. Lough *et al.* [101] postulated a non-virion movement complex of vRNA, CP and TGB1, based on the finding that a C-terminal CP truncation, which maintained the ability to encapsidate the vRNA, nevertheless blocked viral movement, i.e. encapsidation of vRNA alone was not sufficient for transport. On the other hand, filamentous particles were observed in PD of PVX-infected plants and immuno-labelled with antibodies specific to CP in assembled virions, pointing towards assembled virions as the mobile form [70]. Subsequent research showing that the C-terminus of CP is required for interaction with TGB1 at the 5' end of virions [19,102,103] resolved this apparent discrepancy, suggesting that the movement complex may consist of TGB1-bound virions [54,102]. In agreement with this, CP localises to PD only during infection, not through virus-free co-expression with TGB proteins, suggesting that it is recruited only as a vRNA-containing virion [54]. Consequently, the inability of PUM-BiFC to detect PVX vRNA inside PD was interpreted as inaccessibility of the vRNA within the particle [54]. The recruitment of Csy4*-GFP by PVX inside PD now shows that the vRNA can in fact remain accessible.

PVX vRNA inside the virion remains vulnerable to ~14 kDa RNaseA and ~12 kDa RNase T1 [104,105]. Thus, it is conceivable that the 15 kDa Csy4* domain of Csy*-GFP could potentially access the vRNA within PVX particles. However, Csy4* binds a stem-loop secondary structure and its RNA affinity is mainly dependent on interactions with the dsRNA stem [60], which would be unfolded within the virus capsid [103]. Therefore, Csy4*-GFP recruitment to PD during PVX infection must be due to folded stem-loops. This supports a model in which the movement complex consists of only partially encapsidated vRNA ('single-tailed particles') [54,102] resulting from the first, spontaneous stage of PVX encapsidation proceeding in a 5'→3' direction [15,50,102]. The unencapsidated 3' region of such single-tailed particles

would be accessible to Csy4* binding and retain folded secondary structures such as *csy* stem-loops. It remains possible that Csy4* binding artificially stabilises the stem-loops and thereby prevents full encapsidation that might occur in a native PVX infection. However, increasing the number of stem-loops from 2x to 12x had a negative effect on virus spread in the absence of Csy4*-GFP, which could be due to stem-loops remaining folded during transport, independent of stabilisation by Csy4. Whilst we cannot address the situation in a native PVX infection with our artificially tagged vRNA system, Csy4*-GFP recruitment to PD does at minimum demonstrate that full encapsidation and unfolding of secondary structures are not requirements for PVX RNA to enter PD.

In the same way, the ability of Csy4* to detect TMV vRNA and PVX.ΔTGB.30k vRNA inside PD indicates that secondary structures can remain present in vRNA during 30k-mediated virus transport. Based on complexes of 30k and vRNA formed *in vitro* it has been suggested that cooperative binding of 30k protein unfolds vRNA to enable passage through PD [106,107], and the 30k-vRNA complex is inaccessible to ribosomes [108]. However, Citovsky *et al.* [106] also suggested that double-stranded regions of the TMV vRNA would remain structured within the 30k movement complex based on the selectivity of 30k for single-stranded nucleic acids and its inability to unwind dsDNA *in vitro* [106,107], an assumption now confirmed by Csy4*-GFP and in agreement with experiments investigating 30k-mediated transport of its own mRNA [57, 58]. This also supports suggestions that the TMV replicase may be co-transported through PD along with the vRNA, enabling a faster initiation of replication in secondary infected cells [13]. TMV replicase recognises secondary structures [14, 109] which may remain folded just like the *csy* tags.

Related to this, the PVX.ΔTGB.30k.4x*csy*.RFP-2A-CP movement-complemented virus also showed that PVX VRCs containing vRNA and replicase can be recruited to PD-adjacent structures in the absence of the native PVX movement proteins. The link between the ectopic, PD-targeting 30k protein and the PVX replicase is likely either provided by the vRNA, or perhaps by host proteins bridging the unrelated viral proteins, as 30k was unable to recruit the PVX replicase (or CP) directly in the absence of infection. This is similar to the situation in native PVX, where the TGB proteins alone do not recruit replicase or CP in the absence of infection [54,73] and TGB2 probably interacts with RNA as well as the replicase [73,110]. As the PVX replicase likely also binds to RNA secondary structures [111], the accessibility of PVX vRNA secondary structures during both, TGB- and 30k-mediated vRNA transport, and the infection-dependent recruitment of the replicase suggests that the PVX replicase could potentially be co-transported as well. Such a mechanism could generally increase the speed and efficiency of plant virus cell-to-cell movement [9,112]. Our data do not provide any insights into whether 30k trafficked to PD in association with PVX VRCs, or 'captured' mobile VRCs after targeting the channels. However, despite its lack of sequence-specificity [106], 30k protein bound to its own mRNA and transported it to PD [57, 58], indicating it may associate with nearby RNA co-translationally. Recruitment of viral replicases by movement complexes could therefore also increase the specificity of movement protein-mediated RNA transport.

Co-translational association of virus-expressed proteins with vRNA is also suggested by the preferential access that Csy4*-GFP appears to have when expressed *in cis* compared with *in trans*. In the untagged 'self-tracking' PVX.Csy4*-GFP-T2A-CP construct, recruitment of Csy4*-GFP to vRNA within PD was noticeably stronger than when Csy4*-GFP was co-expressed with an untagged virus by agroinfiltration. Furthermore, spread of tagged 'self-tracking' PVX.2x*csy*.Csy4*-GFP-T2A-CP and PVX.Csy4*-GFP-T2A-CP.2x*csy* constructs was severely restricted whilst co-expression of Csy4*-GFP with 2x*csy*-tagged virus *in trans* had a far less dramatic effect.

From our data we currently cannot distinguish if impaired spread was due to a reduction in replication, cell-to-cell movement, or both. However, the presence of typical VRCs with vRNA 'whorls' did not suggest any fundamental changes to replication. The *csy* stem-loops were also located far downstream of the origin of assembly [15] and thus unlikely to have affected encapsidation, and Csy4*-GFP access to secondary structures within PD-transported vRNA suggests that full encapsidation is not required for movement anyway. If Csy4-binding sterically hindered vRNA passage through PD, the severity of impairment might be proportional to the fraction of genomes bound, with *cis*-expression of Csy4*-GFP perhaps allowing preferential access, particularly if progeny RNA is directly inserted into PD from adjacent membranous VRC compartments [54]. At the same time, PD labelling by Csy4*-GFP may not be directly proportional to movement efficiency, as PVX RNA probably continues to accumulate in PD even after cell-to-cell movement. PVX continuously produces TGB1 and CP for at least 96 h after infection [113] and we have previously observed that PD accumulation of CP continues to increase behind the leading edge of PVX infections [54]. Thus, strong PD labelling by Csy4*-GFP could result either from the majority of the RNA being bound (and getting 'stuck'), or a smaller fraction of bound RNA gradually accumulating. On the other hand, sterical hindrance may not be the main or only factor, as TMV 30k co-transported a BglG RNA reporter through PD with an mRNA tagged with 18 cognate stem-loops [58], presumably a much stronger sterical hindrance (but also a different movement system). It is also unclear how the effect of the tagging position (Fig 7F, 7I, 7M and 7N) would relate to sterical hindrance, though it might affect the ratio of Csy4*-GFP bound subgenomic RNAs not contributing to movement. Notably, in the most severely restricted 'self-tracking' construct, PVX.2x*csy*.Csy4*-GFP-T2A-CP, PD labelling was only observed (Fig 7H) in the rare cases when small multi-cellular lesions had formed, which might indicate that movement was impaired before RNA reached PD. For now, these possibilities remain speculative. But regardless of the mechanism, the impairment caused by expressing Csy4*-GFP *in cis* rather than *in trans* fits best with a model where translation of viral gene products occurs in close spatial proximity with replication.

Whilst Csy4*-GFP has proven to be a versatile system for *in vivo* imaging of viral RNA genomes, it has yet to be used for visualisation of cellular mRNAs, for which it would be useful to identify the peroxisome-adjacent granules labelled by Csy4*-GFP. On the other hand, Csy4*-based RNA imaging likely has potential beyond plant biology.

## Materials & Methods

### Cloning procedures

See Supplemental S1 Table for all primer sequences. The cloning of GFP-fused Csy4* reporter constructs, and the tagging of the PVX genome have been partially described elsewhere [114]. To produce Csy4* reporters, a synthetic DNA fragment with Gateway attB sites at each end and containing a *Pseudomonas aeruginosa* UCBPP-PA14 Csy4 ORF (https://www.ncbi.nlm.nih.gov/nuccore/NC_008463.1?from=2927517&to2928080) with a H29A mutation and codon-optimised based on *Nicotiana benthamiana* codon usage (http://www.kazusa.or.jp/codon/cgi-bin/showcodon.cgi?species=4100) [115], followed in-frame by an SV40 NLS [116] flanked by two *Kpn*I sites, and a stop codon flanked by two *Age*I sites, was recombined into the Gateway entry vector pDONR201 (LifeTechnologies). Either the NLS, the stop codon, or both were then removed by sequential restriction digests and re-ligation, resulting in Csy4*-NLS-stop, Csy4*-NLS, Csy4*-stop or Csy4* cassettes, respectively (confirmed by sequencing). These entry clones were then recombined with the destination vectors pGWB405 or 406, respectively [117], to produce expression vectors encoding the fusion proteins

GFP-Csy4*-NLS, Csy4*-NLS-GFP, GFP-Csy4* and Csy4*-GFP. The cellular markers DCP1-RFP, RDR6-RFP, RFP-RBP47 and RFP-SRL [64–66] have been previously described.

To tag the PVX genome, a synthetic DNA fragment, consisting of two *csy* stem-loops separated by an *Eco*RI site and flanked by paired *Not*I, *Pac*I and *Nhe*I sites, was inserted into the unique *Nhe*I site preceding the RFP-2A-CP ORF in a previously published replicon vector containing the PVX genome under control of a cauliflower mosaic virus 35S promoter in a pTRAc binary vector (PVX.RFP-2A-CP) [118]. The number and orientation of tags was random and clones with the desired number of tags were identified by sequencing. To insert *csy* stem-loops downstream of CP in 'self-tracking' PVX constructs, a *Not*I site was generated immediately following the CP stop codon by PCR mutagenesis.

To construct cytoplasmic MS2CP-GFP and λN$_{22}$-GFP fusions, previously generated Gateway entry clones containing ORFs encoding either MS2CP [33,55] (lacking the C-terminal three codons to prevent virion assembly, and containing a V29I mutation to increase RNA affinity [119,120]) or λN$_{22}$ [34] fused to GFPc3 [121] monomerised by an A206K mutation [122] were recombined with pGWB402Ω [117], for expression from a 35S promoter.

TMV genomes tagged with binding sites for different RNA imaging systems were generated based on the replicon vector pTRBO [123], which contains a TMV genome under control of a 35S promoter and replaces the CP ORF with a *Pac*I-*Not*I cloning site. pTRBO-based TMV.ΔCP.RFP has been described previously [38]. RFP was amplified without or with two downstream binding sites (2x*ms2* or 2x*boxB*, respectively), and flanking 5' *Pac*I and 3' *Not*I sites as well as Gateway attB adapters by overlap extension PCR. Final PCR products were cloned into pDONR201 by Gateway recombination. After sequence verification, cassettes were excised with *Pac*I/*Not*I and ligated into pTRBO linearised with the same restriction sites, yielding TMV.ΔCP.RFP.2x*ms2* and TMV.ΔCP.RFP.2x*boxB*, respectively. TMV.ΔCP.RFP was tagged with 2x*csy* using the synthetic DNA fragment described above inserted into the unique *Pac*I site, yielding TMV.ΔCP.RFP.2x*csy*. For imaging with PUM-BiFC system, the untagged TMV.ΔCP.RFP. construct was used, as the engineered Pumilio homology domains PUMHD3794 and PUMD3809 recognise the TMV genome without requirement for any tags [38].

To replace the PVX TGB movement proteins with TMV 30k, the 30k ORF and a 4x*csy*.RFP-2A-CP cassette from a previously generated PVX.4x*csy*.RFP-2A-CP replicon were PCR-amplified with 25 bp overlaps to a PVX.GFP-2A-CP vector [124]. The vector backbone was also PCR-amplified and the three fragments were assembled with NEBuilder HiFi DNA assembly kit (New England Biolabs) according to manufacturer's instructions. The fragment boundaries were selected such that the start codon of TMV 30k exactly replaced the start codon of PVX TGB1, and the 30k stop codon replaced the PVX TGB3 stop codon.

To generate a 'self-tracking' PVX construct, Csy4*-GFP was amplified from the previously generated expression vector with primers that introduced flanking *Nhe*I and *Kpn*2I sites. The PCR product and a pTRAc-based PVX replicon construct in which the foot and mouth disease virus 2A sequence had been replaced with a thosea asigna virus 2A (T2A) sequence [80] were both digested with *Nhe*I and *Kpn*2I and ligated to produce PVX.Csy4*-GFP-T2A-CP. Due to the presence of the T2A peptide, translation of the fused ORF results predominantly in separate Csy4*-GFP-T2A and CP polypeptides [80]. *Csy* stem-loops were then introduced as above by ligation into the *Nhe*I site, resulting in PVX.2x*csy*.Csy4*-GFP-T2A-CP, or into a *Not*I site introduced downstream of the *CP* ORF, resulting in PVX.Csy*-GFP-T2A-CP.2x*csy*.

## Plant inoculations

Four-week old *N. benthamiana* plants grown on soil under natural light were used for all experiments. Infectious virus constructs were inoculated by microprojectile bombardment

using a custom-built gene gun [125]. 0.3 to 2.5 µg plasmid DNA were mixed thoroughly with 50 µL 1 µm gold carriers (BioRad) (40 mg/mL) in 50% glycerol. 50 µL 2.5 M $CaCl_2$ and 20 µL 0.1 M spermidine were added and again mixed thoroughly. After incubation on ice for 1 min, gold particles were sedimented for 30 s at 5000 rpm in a benchtop centrifuge, washed with 250 µL ethanol and then resuspended in 30 µL ethanol. For comparisons of lesion sizes produced by tagged and untagged PVX, plasmid amounts were standardised to 1.45 µg DNA/50 µL gold carriers to ensure comparable inoculum doses. 2.5-5 µL of DNA/gold mixture were applied to a 13 mm Swinnex Filter Holder (MerckMillipore), which was screwed onto the gene gun nozzle. Gold particles were shot onto *N. benthamiana* leaves from ~2 cm distance at a $N_2$ pressure of 20 – 25 psi.

Csy4* fusions and other reporter constructs were transiently expressed by agroinfiltration. Strain AGL1 agrobacteria were electroporated with the relevant binary plasmids and grown at 28°C for 2 – 3 d on double selection plates containing rifampicin and the appropriate antibiotic for plasmid selection, either spectinomycin or kanamycin, respectively. Colonies were verified by PCR, grown in double selection liquid culture at 28°C, 220 rpm for 2 d and then stored at -70°C as 30% glycerol stocks. Agrobacteria were streaked from glycerol stocks onto double selection plates and grown at 28°C for 2 days. Plates were kept in the fridge and used for 1 – 2 weeks. For agroinfiltration, agrobacteria were inoculated from re-streak plates into single selection liquid media containing only the plasmid-selective antibiotic but no rifampicin, and grown overnight at 28°C, 220 rpm. Cells were pelleted, and resuspended in infiltration medium (10 mM morpholino ethanesulfonic acid (MES), 10 mM $MgCl_2$, 15 µM acetosyringone). After 1 h incubation in the dark, agrobacterium suspensions were diluted to an optical density at 600 nm ($OD_{600}$) of 0.1 for each individual expression plasmid-carrying strain. Infiltration was via the abaxial surface using a needle-less syringe pressed onto a small incision made with a syringe needle. For co-localisation with virus infections, agroinfiltrations were carried out 2-3 days after virus inoculation. The ectopic silencing suppressor 19k from *Tomato bushy stunt virus* (TBSV)[126] was co-expressed with Csy4* constructs to compensate for suppression of agrobacterium-mediated expression in the presence of PVX [126].

## Imaging and image processing and analysis

Inoculated leaves were imaged at 3 – 14 d after virus inoculation and 3 – 4 d after agroinfiltration. Leaves were detached and immobilised on glass slides with double sided sticky tape, lower epidermis facing up for agroinfiltrated tissues, bombarded side facing up for virus-inoculated samples. For staining of PD-associated callose, 0.1 mg/ml aniline blue solution in water was infiltrated into leaves immediately before imaging. Imaging was carried out on a Zeiss 710Meta confocal laser scanning microscope mounted on an upright Axio Imager Z2 stand (Carl Zeiss, Jena). 2.5x, 10x and 20x long-distance air lenses were used for imaging whole infection sites and for overview images. A 40x water dipping lens immersed in a drop of water placed directly on the plant sample was used for subcellular localisations. Additional imaging was carried out on a Nikon A1R confocal laser scanning microscope on an upright Nikon NiE base (Nikon UK, Kingston upon Thames) equipped with 10x long-distance air and 40x and 60x water dipping lenses. Excitation/detection wavelengths were 488/499 – 530 nm for GFP/YFP, 561/587 – 626 nm for RFP and 405/453-499 nm for CFP/aniline blue, respectively. Chlorophyll autofluorescence was detected at 650-690 nm. GFP and RFP channels were imaged sequentially to minimize bleed-through. Images were collected using microscope proprietary Zen software (Carl Zeiss, Jena) and exported as .TIF files for processing. Figure panels were assembled using Powerpoint (Microsoft) and Photoshop (Adobe). All brightness and contrast adjustments were applied to whole images.

To quantify viral lesions sizes, Fiji ImageJ2 (https://imagej.net/software/fiji/; [127]) was used for semi-automated image analysis. For single images, maximum z-projections were generated and the colour channels split. Tile scans were stitched together and a maximum projection of the channel of interest prepared in Zen software before exporting single RGB files. All images were pre-processed as follows: Smooth, Despeckle, and Gaussian blur (75, scaled), then 'Yen' Thresholding was applied. To measure fluorescent lesion sizes, Analyze Particles was used (with edges excluded) to select the regions of interest (= lesion) and the area measurements were recorded. For tile scans, to use the same pre-processing and segmentation macro, the 8-bit converter was applied and the appropriate scale set in μm. For statistical analysis, the median of each biological repetition was taken as a single data point. An unpaired Student's t-test was applied for the comparisons of PVX.2x*csy*.RFP-2A-CP with PVX.12x*csy*.RFP-2A-CP, PVX.Csy4\*-GFP-T2A-CP with PVX.2x*csy*.Csy4\*-GFP-T2A-CP and PVX.Csy4\*-GFP-T2A-CP with PVX.Csy4\*-GFP-T2A-CP.2x*csy* with a confidence interval of 0.95. For the analysis of untagged PVX.RFP-2A-CP and PVX.2x*csy*.RFP-2A-CP viruses on tissue expressing either free GFP or Csy4\*-GFP, a two-way ANOVA and Tukey post-hoc test was applied. Statistical analysis was carried out in R.

### RT-PCR

Total RNA was extracted from leaf tissue using a Qiagen RNeasy Plant Mini Kit (Qiagen) according to manufacturer's instructions. First strand cDNA was synthesized using a Lunascript RT SuperMix Kit (New England Biolabs) according to manufacturer's instructions. cDNA was then used as a template for a PCR reaction using GoTaq G2 Flexi DNA Polymerase (Promega) under the following conditions: $T_a$ of 50 °C, 30 cycles, 2 mM $MgCl_2$, 1.5 minute extension, with primers OvercoatF/CP28R. Samples were visualised using ethidium bromide 1% agarose gel electrophoresis.

### Western blotting

Protein was extracted by homogenising 200 mg of leaf material in 400 μL of Laemmli buffer (62.5 mM Tris-HCl pH 6.8, 10% glycerol, 2% lithium dodecyl sulfate (LDS), 0.1% bromophenol blue, 2.5% β-mercaptoethanol) using a micropestle. Proteins were denatured at 70°C for 5 min and solid plant debris sedimented 1 min at 14 krpm in a benchtop centrifuge. 5 μl aliquots were separated alongside PageRuler pre-stained protein size standard (ThermoScientific) on a 12.5% sodium dodecyl polyacrylamide gel (SDS-PAGE) in Tris-glycine buffer. Proteins were electroblotted onto a polyvinylidene fluoride (PVDF) membrane for 3 h at 60 V and detected with anti-GFP primary antibody (1:2000 mouse monoclonal GF28R, Invitrogen), horseradish peroxidase-conjugated rabbit anti-mouse secondary antibody (1:10,000, Sigma) and a 3:1 mixture of Pico and Femto SupersignalWest substrate (ThermoScientific). Chemiluminescence was recorded on a G:BOX Chemi XT4 (Syngene) and exported as.TIF files.

### Supporting information

**S1 Fig. Localisation of additional GFP fusions of Csy4\*.** (A) N-terminal GFP fusion with nuclear localisation signal (NLS) shows exclusively nuclear localisation. Inset shows enrichment in nucleolus (arrow). (B) C-terminal GFP fusion with NLS shows exclusively nuclear localisation. Inset shows enrichment in nucleolus (arrow). (C) N-terminal GFP fusion without a NLS shows protein aggregates. All images are maximum intensity z-projections. GFP fluorescence shown green, chlorophyll auto-fluorescence shown blue. Scale bars, 10 μm. (TIF)

**S2 Fig. Co-localisation of Csy4\*-GFP with different RNA granule markers.** (A-C) Co-localisation with P-body marker DCP1. Arrow heads indicate DCP1 granules co-localised with Csy4\*-GFP. (D-F) Co-localisation with stress granule marker RBP47. (G-I) Co-localisation with tasiRNA processing body marker RDR6. Arrow heads indicate RDR6 granules co-localised with Csy4\*-GFP. Left column: GFP channel, middle column: RFP channel, right column: merge. GFP fluorescence shown green, RFP fluorescence shown magenta. All images are maximum intensity z-projections. Scale bars, 10 μm.
(TIF)

**S3 Fig. Csy4\*[mut]-GFP does not label PVX.2xcsy.RFP-2A-CP RNA.** (A-B) Perinuclear VRC. GFP fluorescence is diffuse with no 'whorls' and Csy4\*[mut]-GFP is not enriched in the nucleolus. (C-D). Absence of GFP fluorescence from RFP-2A-CP labelled PD (arrow heads). GFP fluorescence shown green, RFP fluorescence shown magenta. Image in (A,B) is a single z-section; image in (C,D) is a maximum intensity projection of three z-sections. Scale bars, 10 μm.
(TIF)

**S4 Fig. Full images corresponding to main Fig 6E and 6F panels.** (A-B) Full maximum intensity projection of the entire z-stack used in Fig 6E (which is a projection of two z-sections). RFP fluorescence indicating PVX infection is visible in the second cell from the left (arrow heads). (C-D) Full image used for Fig 6F shows a multitude of PD-associated structures containing both RFP-2A-CP and 165k-GFP. Arrows: approximate direction of viral spread. Dashed boxes: regions enlarged in Fig 6E, F, respectively. GFP fluorescence shown green, RFP fluorescence shown magenta, CFP and aniline blue fluorescence shown cyan. All images are maximum intensity z-projections. Scale bars, 10 μm.
(TIF)

**S5 Fig. TMV 30k protein does not directly recruit PVX replicase or CP to PD.** (A-B) Transient co-expression by agroinfiltration of 165k-RFP and 30k-GFP in *N. benthamiana* leaf epidermis. 165k-RFP remains confined to the nucleus with enrichment in the nucleolus (arrow). (C-D) Transient co-expression by agroinfiltration of RFP-CP, unfused CP and 30k-GFP in *N. benthamiana* leaf epidermis. RFP-CP remains nucleo-cytoplasmically distributed with no enrichment at PD or PD-adjacent structures. GFP fluorescence shown green, RFP fluorescence shown magenta. All images are maximum intensity z-projections. Scale bars, 10 μm.
(TIF)

**S6 Fig. Systemic movement of 'self-tracking' PVX constructs.** (A-C) Representative images from systemic leaves of plants infected with untagged PVX.Csy4\*-GFP-T2A-CP (A), upstream-tagged PVX.2x*csy*.Csy4\*-GFP-T2A-CP (B) and downstream-tagged PVX.Csy4\*-GFP-T2A-CP.2x*csy* (C), respectively, at 14 days post inoculation (dpi). GFP fluorescence shown green, chlorophyll autofluorescence shown blue. All images are maximum intensity z-projections. Scale bars, 1 mm. (D) RT-PCR analysis of systemically infected leaves at 14 dpi. A segment of the PVX genome from the end of the TGB3 ORF to the beginning of the CP ORF was amplified. Arrow heads on right indicate expected product size when complete Csy4\*-GFP-T2A-CP ORF is present. 1 to 18: three biological replicates with six plants each. 0x: untagged PVX.Csy4\*-GFP-T2A-CP, 2x [u.s.]: upstream-tagged PVX.2x*csy*.Csy4\*-GFP-T2A-CP.
(TIF)

**S7 Fig. Co-localisations of 'self-tracking' PVX.Csy4\*-GFP-T2A-CP with viral proteins expressed *in trans*.** (A) RFP-TGB2 localises to the VRC, appearing to surround vRNA (arrow

head). (B) RFP-CP localises to the VRC, also surrounding vRNA (arrow heads). (C) 165k-RFP replicase marker localises to granular structures among the vRNA in the VRC. (D, E and G) TGB1-RFP (D), RFP-TGB2 (E) and RFP-CP (G) co-localise with vRNA inside plasmodesmata (arrow heads). (F) TGB3-RFP is located in membrane structures at plasmodesmata entrances, separate from vRNA inside the channels (arrow heads). Top rows, merged images, bottom rows, RFP channel only. GFP fluorescence shown green, RFP fluorescence shown magenta. All images are single z-sections, except (F) which is a maximum intensity projection of two z-sections. Scale bars, 10 μm.
(TIF)

**S8 Fig. Effect of Csy4\*-GFP on PVX spread when expressed *in trans*.** Lesion sizes of untagged (0x) or 2x*csy*-tagged (2x) PVX.RFP-2A-CP at 6 days post inoculation. Virus was inoculated on leaves expressing either unfused GFP or Csy4\*-GFP (2X/Csy4\*-GFP: 85, 2x/GFP: 99, 0x/Csy4\*-GFP: 93, and 0x/GFP: 100 infection sites; n = 3 independent experiments). Two-way ANOVA: The interaction between the two variables, number of tags *versus* expressed GFP construct, was significant (\*\*\*; $p = 1.13 \times 10^{-5}$), however, pairwise Tukey test found no significant differences between treatments.
(TIF)

**S1 Table. Oligonucleotide primers.**
(XLSX)

## Acknowledgements

We thank Ulrich Commandeur and Christina Dickmeis (University of Aachen, Germany) for the pTRAc-based PVX.RFP-2A-CP construct, Nathan Pumplin (Swiss Federal Institute of Technology (ETH-Zürich), Zürich, Switzerland/ Norfolk Healthy Produce, Davis, CA) for the gift of DCP1-RFP and RDR6-RFP markers, Markus Fauth (University of Frankfurt, Germany) for the RFP-RBP47 marker, Aiming Wang (Agriculture and Agri-Food Canada, London, ON, Canada) for SGS3 and RBP47 constructs, and Petra Boevink (James Hutton Institute, Dundee, U.K.) for the RFP-SRL marker.

## Author contributions

**Conceptualization:** Jens TILSNER.

**Data curation:** David Burnett, Mohamed Hussein, Zoe Kathleen Barr, Kathryn M. Wright, Jens TILSNER.

**Formal analysis:** David Burnett, Mohamed Hussein, Zoe Kathleen Barr, Jens TILSNER.

**Funding acquisition:** Jens TILSNER.

**Investigation:** David Burnett, Mohamed Hussein, Laura Newsha Näther, Kathryn M. Wright, Jens TILSNER.

**Methodology:** David Burnett, Mohamed Hussein, Zoe Kathleen Barr, Jens TILSNER.

**Project administration:** Jens TILSNER.

**Resources:** Zoe Kathleen Barr, Jens TILSNER.

**Supervision:** Jens TILSNER.

**Validation:** David Burnett, Mohamed Hussein, Zoe Kathleen Barr, Jens TILSNER.

**Visualization:** David Burnett, Mohamed Hussein, Zoe Kathleen Barr, Laura Newsha Näther, Kathryn M. Wright, Jens TILSNER.

**Writing – original draft:** Jens TILSNER.

**Writing – review & editing:** David Burnett, Mohamed Hussein, Zoe Kathleen Barr, Laura Newsha Näther, Kathryn M. Wright, Jens TILSNER.

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
