## [Decision Letter · Decision Letter 0]

4 Sep 2024

Dear Dr Jens Tilsner,

Thank you very much for submitting your manuscript " Live-cell RNA imaging with the inactivated endonuclease Csy4 enables new insights into plant virus transport through plasmodesmata" (PPATHOGENS-D-24-01373) for review by PLOS Pathogens.

Your manuscript was fully evaluated at the editorial level and by three independent peer reviewers. The reviewers appreciated the attention to an important problem, but raised some substantial concerns about the manuscript as it currently stands. These issues must be addressed before we would be willing to consider a revised version of your study. We cannot, of course, promise publication at that time. We therefore ask you to modify the manuscript according to the review recommendations before we can consider your manuscript for acceptance. Your revisions should address the specific points made by each reviewer which all came to the conclusions that the major claims are not sufficiently supported by the provided experimental data. Also, the manuscript suffers from being descriptive and rather methodological without providing any novel (mechanistic) insights into cell-to-cell movement of plant RNA viruses.

I am returning your manuscript with three reviews. The reviewers came to nearly the same conclusions about the paper, as you will see. After reading the reviews and looking at the manuscript, I agree with the criticisms raised and recommend Major Revision. I am sorry I cannot be more positive at the moment, however we are looking forward to receiving your revision. With a lot of work, the manuscript will be suitable for a resubmission, if you so wish to do so. In this respect, a number of relevant additional experiments are suggested in particular by reviewer 2 and 3 (see major issues), that should be carefully addressed before considering resubmission.

We cannot make any decision about publication until we have seen the revised manuscript and your response to the reviewers' comments. Your revised manuscript is also likely to be sent to reviewers for further evaluation.

Sincerely,

Christophe Ritzenthaler

Academic Editor

PLOS Pathogens

Savithramma Dinesh-Kumar

Section Editor

PLOS Pathogens

Michael Malim

Editor-in-Chief

PLOS Pathogens

orcid.org/0000-0002-7699-2064

Reviewer's Responses to Questions

**Part I - Summary**

Reviewer #1: The manuscript by Burnett et al. described a new method for labeling viral genomic RNA based on the CRISPR-associated endonuclease Csy4. Although several other similar systems have been reported, this Csy4-based labeling system seems to be more sensitive possible due to its high affinity for its cognate stem-loop. Apart from this RNA labeling system, the results reported in the manuscript have not, in my opinion, yielded new insights concerning the cell-to-cell movement of plant RNA viruses.

Reviewer #2: This article presents an interesting piece of work in which the authors address a question that has remained elusive for a long time in plant virology: what is the physical entity that is transported through plasmodesmata? To do so, the authors use the high affinity of an inactivated bacterial endonuclease for its structured RNA motif.

Reviewer #3: The manuscript “Live-cell RNA imaging with the inactivated endonuclease Csy4 enables new insights into plant virus transport through plasmodesmata” by David Burnett and Mohamed Hussein et al. uses the inactivated endonuclease Csy4 to visualize viral RNA in living plant cells.

Unlike other live cell RNA imaging systems published before, this manuscript for the first time showns vRNA labeling inside plasmodesmata (PD) during virus infection. Based on this observation a range of conclusions are drawn and discussed. Among the conclusions is that folded RNA regions may still be present during transport through PD, PVX RNA may not be fully encapsidated during transport and replication proteins may be co-transported within the vRNP through the PD pore. Movement complementation experiments using the tobacco mosaic virus (TMV) movement protein (MP) are employed to demonstrate that also in the presence of the TMV MP PVX replication complexes are found close to PD. Finally the authors engineered a self-tracking virus expressing Csy4* and the cognate stem loops. While Csy4*-GFP expressing virus moved between cells, movement of the virus containing the stem loops was impaired.

The main novelties of this study are the application of the Csy4* system to label RNA in plant cells and the labeling of RNA inside the PD pore during virus infection. Consequently, the discussion of the potential implications of this finding for the mechanism of viral RNA movement and the nature and structure of the moving RNA-protein complex is a key point of the manuscript. Given the predominantly descriptive data obtained by confocal microscopy, the conclusions are naturally inferred from the observations. Thus, while it is certainly important to include possible conclusions drawn from the observations presented here on vRNA movement through PD and virus replication into current models, care must be taken not to over-interpret the observations.

**Part II – Major Issues: Key Experiments Required for Acceptance**

Reviewer #1: (1) The idea that PVX RNA is not fully encapsidated during movement is possible but not new, also drawing such a conclusion based on the observation that PVX RNA granules were observed in the plasmodesmata is not sufficient. It is possible that Csy4-GFP binds to PVX RNA before it enters the PD, and its high binding affinity prevents its release from the RNA in the plasmodesmata channel. A direct comparison of the cell-to-cell movement efficiency between the combinations PVX.4xcys.RFP-2A-CP and PVX.4xcys.RFP-2A-CP + Csy4*-GFP may be helpful.

(2) The claim that the recruitment of VRC to PD is not virus-specific is also not well supported: Considering the cell-to-cell movement of PVX.ΔTGB.30k.4xcsy.RFP-2A-CP is slower than wild-type PVX (line 366), and I guess the percentage of PD that is associated with VRC in the cell infected by PVX.ΔTGB.30k.4xcsy.RFP-2A-CP should be lower than that in the cells infected by PVX.4xcsy.RFP-2A-CP. However, there is no such data in the paper. Fig.6F could be due to VRCs moving with the cell stream are hung up by the TMV MP multimers protruding at the PD, which allow the transport of viral RNA at a low efficiency. This is actually consistent with the previous results that the interaction between TGB2 and RdRp is required for efficiently recruiting VRC to PD (see doi: 10.1128/JVI.01635-18). Alternatively, it is possible that TMV MP could interact with PVX RdRp or CP via a common yet unknown mechanism to recruit PVX VRC. The authors need to exclude all these possibilities.

Reviewer #2: The work is well done and the article well written, although my main objection is that the conclusions that the authors draw from the results are not sufficiently supported by them. Even recognizing that the work is done with sufficient detail and meticulousness, there are three clear objections to it that make it require important modifications. i) the authors describe as new the highly sensitive RNA live-cell reporter based on an enzymatically inactive form of the small bacterial endonuclease Csy4, which binds to its cognate stem-loop with nanomolar affinity, but in fact they had already described it previously in a methodological article with a great level of detail (Burnett et al., 2020). In fact, in that article they already state that the inclusion of two tags is sufficient for a good image and here they use in several experiments constructions with 4 tags.; ii) I think the statement in the results section 'Csy4* labels viral RNA inside plasmodesmata' is too excessive and lacks conclusive evidence. To claim that when the signal of Csy4*-GFP concentrated in the inter-cellular cell wall space is clearly seen means that it is inside plasmodesmata seems to me to be weak. To claim that it is 'inside' and not 'associated' would require resolution at the level of electron microscopy; iii) It is difficult to assimilate and interpret that the construction untagged PVX.RFP-2A-CP rendered signicant PD labeling when compared with the tagged one; Indeed authors stated that the strong recruitment of Csy4*-GFP by PVX RNA even without csy tags facilitated co-localizations with transiently expressed PVX proteins. This suggests to me that the effect is being unspecific. Finally, authors speculate that the ability of Csy4* to detect TMV vRNA and PVX.�TGB.30k vRNA inside PD indicates that secondary structures remain present in vRNA during 30k-mediated virus transport. Based on complexes of 30k and vRNA formed in vitro it has been suggested that cooperative binding of 30k protein unfolds vRNA to enable passage through PD, and the 30k-vRNA complex is inaccessible to ribosomes. To demonstrate that Csy4* is accessible to the structured RNA an easy experiment would be to carry out EMSA assays of the corresponding construct and Csy4* in the presence and absence of TMV MP.

Finally, one of the main conclusions is that the ability of Csy4* to label vRNA inside PD may be due to its higher affinity for its cognate target sequence compared with the other imaging systems. However, it is said that tight binding of Csy4*-GFP interfered with access to vRNA during replication or recruitment for movement. Both affirmations are difficult to reconcile.

Reviewer #3: In my opinion, the main concern about the data is that it is not clearly shown that indeed genomic RNA is seen in PD, or if rather the subgenomic RNAs expressed during virus infection are bound by the Csy4*-protein when expressed in trans or from the virus. The conclusions drawn on virus movement are based on the assumption that genomic viral RNA is labelled. However, the stem loops and the Csy4*-GFP coding sequence are located in a region, where a subgenomic RNA is made (see ref 109).

Have the authors tried to experimentally determine whether genomic or subgenomic RNAs are preferentially bound? I imagine that an experimental approach to demonstrate which RNAs are bound by the protein is very difficult, as it would require e.g. immunoprecipitations of minute amounts of the Csy4*-GFP-expressing virus from infection sites and the determination of the nature of the bound RNAs. However, these experiments could also give an answer to the question, which proteins are in the complex. By contrast, as the tagged virus PVX.2csy.RFP-2A-CP moves systemically, agroinfiltrated Csy4*-GFP could be used in systemic tissue to immunoprecipitate the CSY4*-GFP-RNA complex and determine, whether genomic or subgenomic RNAs are preferentially bound.

The data in Figure 1D indicate that Csy4*-GFP also binds to cellular RNA in the absence of the cognate stem loops (RNA-rich nucleolus and granules). In Fig. 2D, 3E, Fig. 7 A-D it is shown that Csy4*-GFP efficiently binds to vRNA in the absence of stem loops. However these findings are not sufficiently discussed. I would have liked to see a discussion on the specificity of binding to RNA other than the target RNA or the occurrence of the specific RNA sequences resembling the cognate stem loops in viral and plant cellular RNA. The data in Fig 1 F Csy4*[mut]-GFP are interesting, as this indicates absence of localization to RNA-containing subcellular structures for this mutant. Why was this mutant not employed as control for experiments with viral RNA? It would be nice to see the absence of signal in PD when this mutant is co-expressed with virus containing stem loops, complementary to the data shown in Fig. 3 C and D. Moreover, how do the authors interpret the results in Fig. 3 E,F? In line 254 – 256 the authors say that no Csy4*-GFP signal in PD was seen in the absence of virus infection. So I suppose the authors conclude that the PD labeling of Csy4*-GFP during PVX infection without stem loops results from RNA labeling of viral RNA inside PD. Why would endogenous RNA moving through PD not be labeled? Do the authors think this is because of high concentration of virus RNA inside the PD or maybe because viruses dilate PD and only dilated PD can accommodate the Csy4*-GFP inside the PD pore?

Line 394: “vRNA was also observed inside PD but with no co-localisation of RFP-CP (Fig 6D, E), unlike native PVX but as expected for movement mediated by TMV 30k” The authors use this difference to the PVX-based observation to demonstrate that the Csy4* system is able to confirm the postulated and previously reported movement mechanisms for PVX (with CP) vs. TMV (without CP). However, the authors do not discuss the faint RFP-2A-CP signal inside the PD in the Figure 6D. Moreover, in Figure 6E, showing the cell at the leading edge of infection I am not convinced that the cell, into which the virus moves, expressed the RFP-2A-CP protein.

Line 484 to 489: This paragraph and the Figure S4 is unclear to me. The difference in lesion size between tagged virus in Csy4*-GFP expressing tissue compared to the other conditions is described as insignificant, but the infection sites according to the box plot are approximately three fold smaller. Fig. S4 legend says that ANOVA found a significant difference. Agroinfiltration appears to affect lesion size, as lesion size of the two-times tagged construct in Figure 5C was approximately two-fold the size of lesions of untagged or tagged virus in the presence of GFP. Could it be that the binding of the vRNA by the Csy4*-GFP transiently expressed in cells ahead of the infection front hinders spread of infection or replication for the condition, where the tagged virus is analyzed?

Fig. 4 and discussion about the sensitivity of the different RNA labeling systems lines 547 to 553. Indeed, as is discussed, it seems likely that the other systems should be able to detect RNA inside PD during virus infection and that the lack of PD labeling by the other systems may be related to the sensitivity, which could be increased when more stem loops are used. However, I would like to mention that the cells shown in B display much less cytoplasmic labeling of the RFP protein compared to the cells in D,F and H. The data as shown might therefore suggest that B focusses at a cell more at the leading edge of infection, where RNA movement occurs, compared to the other infection sites, which look as if both neighboring cells are already fully infected. Maybe the legend should state that cells at comparable positions inside lesions were analyzed. Additional discussion would be needed to explain why there should be a higher sensitivity of the other systems to label RNA at PD upon ectopic expression of the RNA and proteins (references 54 and 55) vs. expression during infection.

**Part III – Minor Issues: Editorial and Data Presentation Modifications**

Reviewer #1: (1) Fig. 4: The panels D, F, and H are not typical as there is no plasmodesmata or VRCs can be found.

(2) Fig. 5 and cognate context: The size of the bands of PVX.12xcys.RFP-2A-CP in Fig. 5G is not equal (lanes 3, 9, and 14). Also, lane 10 contains two bands. Therefore, too much hairpin is genetically unstable, in my opinion. Thus, it needs very carefully consideration to draw the conclusion that multiple csy tags have only a moderate negative effect on viral infectivity.

Reviewer #2: Line 27, abstract: it is said “..movement proteins which bind nucleic acids and target and dilate plasmodesmata”. This is a generalization of the MPs since not all of them have nucleic acid binding properties (e.g MPs of GFLV, CPMV, CLRV, etc..)

Lines 30-32, abstract; “Here, we describe a new, highly sensitive RNA live-cell reporter based on an enzymatically inactive form of the small bacterial endonuclease Csy4, which binds to its cognate stem-loop with nanomolar affinity”. See general comments above.

Line 67: or multiple MPs, nevertheless, some common features have emerged: 1) nucleic acid binding.

See above. Not all MPs bind nucleic acids. Most MPs forming tubular structures have not the capability to bind n.a.

Line 81, when stating that capsid proteins (CPs) often preferentially interact with secondary structures in their cognate viral (v)RNA it is surprising that the paradigmatic and best studied example of AMV (or ilarviruses)-3’UTR was not cited.

Line 86, many MPs are also able to non-specifically transport unrelated viruses.

It seems that some of the works that better describe this scenario have been forgotten.

lines 160 and following; The description of Figure 1 is inappropriate. Figures 1B and 1D appear before 1A and 1C in the text. Modify the figure or change the order of description in the text. The same applies to Figure 4

Lettering of Figure S1 does not match what is described in the legend. Figures are referenced as A to I whereas legend is only referenced for rows (A,B,C).

Line 212 I wonder why authors use Potato virus X tagged with four csy stem-loops if they already showed that two csy stem-loops are sufficient for imaging as described in their methods paper (Burnett et al 2020)

Line 215. … at medium to late infection stages… Please, be more specific in this context. Readers could not remember how dpi are you talking about.

Line 243. however even in the absence of cognate stem-loops in the viral genome, the Csy4*-GFP reporter still weakly showed the characteristic localisation to ‘whorls’ (Fig 2D). This seems quite anomalous. See my general comments

Line 248 I think the statement in the results section ' Csy4* labels viral RNA inside plasmodesmata' is too excessive and lacks conclusive evidence. To claim that when the signal of Csy4*-GFP concentrated in the inter-cellular cell wall space is clearly seen means that it is inside plasmodesmata seems to me to be weak. To claim that it is 'inside' and not 'associated' would require resolution at the level of electron microscopy.

255 When untagged PVX.RFP-2A-CP was used for infection, only very faint PD labelling with Csy4*-GFP was detectable (Fig. 3E,F). I wouldn’t day that. Again, same concern as previous comments.

Lines 267-8. One of ‘only’ must be removed

Line 284, λN22

Lines 291-2 Authors stated that only Csy4*-GFP could detect vRNA in plasmodesmata, indicating that it is the most sensitive amongst these RNA imaging systems, but I think it is also possible to observe PD with λN22.

383, 392, 165k-GFP replicase marker located in ‘caps’ at the entrances of plasmodesmata labelled with aniline blue-staining of callose.

How are the authors sure that this is the PD entry?

396-8. The recruitment of PVX replication complexes by TMV MP would seem to be expected given that Sambade et al had already demonstrated that MP:mRFP as well as its transcript accumulate in PD suggesting that MP has the ability to target its own transcript to the channel. Thus, the targeting of RNA molecules to PD rather represents a sequence-non-specific function of the MP protein because MP has sequence-non-specific nucleic acid-binding activity and is known to facilitate the intercellular spread of other RNA molecules, such as the genomes of unrelated RNA viruses and RNA molecules implicated in the spread of RNA silencing .

488 whilst Csy4*-GFP binding to csy stem-loops seems to have a negative impact on lesion size, it is only the specific combination of viral Csy4*-GFP-T2A expression in cis from a tagged vRNA that severely disrupts viral spread. A ‘self-tracking’ PVX construct demonstrates preferential access to vRNA when Csy4*-GFP is expressed in cis

How can it be stated preferential access to vRNA when Csy4*-GFP is expressed in cis if the untagged vRNA has similar signal intensity?.

Reviewer #3: Line 159: revise sentence. I guess it should be either “each” or “both”

Line 161: Fig 4A should be mentioned in the text before Fig 4B or the order in the Figure changed.

Line 163: Fig 4A to C should be mentioned in the text before Fig 4D or the order in the Figure changed.

Fig. 1: The results of the N-terminally tagged Csy4* and of the nuclear targeted Csy4* shown in Fig 1 A,B, and C are not relevant to the conclusions taken and could be presented as supplementary data.

Fig. 2: Why is there only such a weak magenta signal for the CP in C? I would expect a signal similar to the one shown in Fig. 2A in 2C. Do the two stem loops avoid localization of the CP in the X-body or is packaging inhibited by the two stem loops?

Line 284: the “Lambda” symbol is not shown properly.

Lesion size experiments: It is unclear how many lesions were analyzed in the different experiments

Line 420: Fig. 7 A-J should be mentioned before Fig. 7K or the Figure should be restructured.

Line 433: the representative infection site in E shows only one cell. Maybe this should be mentioned in the legend.

Fig. 7 M and N: The quantified lesion sizes in M and N for the non-tagged virus are different in size, approximately 2 mm2 in (M) vs. 1 mm2 in (N). Why is that? Were the lesions analyzed in different leaf ages or at different times after infection? I did not find the corresponding information.

PLOS authors have the option to publish the peer review history of their article (what does this mean? ). If published, this will include your full peer review and any attached files.

**Do you want your identity to be public for this peer review?** For information about this choice, including consent withdrawal, please see our Privacy Policy .

Reviewer #1: No

Reviewer #2: No

Reviewer #3: No
---

## [Decision Letter · Decision Letter 1]

5 Mar 2025

PPATHOGENS-D-24-01373R1

Live-cell RNA imaging with the inactivated endonuclease Csy4 enables new insights into plant virus transport through plasmodesmata

PLOS Pathogens

Dear Dr. TILSNER,

Thank you for submitting your manuscript to PLOS Pathogens. After careful consideration, we feel that it has merit but does not fully meet PLOS Pathogens's publication criteria as it currently stands. Therefore, we invite you to submit a revised version of the manuscript that addresses the points raised by Reviewer #1.

Please submit your revised manuscript within 30 days May 04 2025 11:59PM. If you will need more time than this to complete your revisions, please reply to this message or contact the journal office at plospathogens@plos.org. Please include the following items when submitting your revised manuscript:

We look forward to receiving your revised manuscript.

Kind regards,

Savithramma P. Dinesh-Kumar

Section Editor

PLOS Pathogens

Sumita Bhaduri-McIntosh

Editor-in-Chief

PLOS Pathogens

orcid.org/0000-0003-2946-9497

Michael Malim

Editor-in-Chief

PLOS Pathogens

orcid.org/0000-0002-7699-2064

**Journal Requirements:**

1) Please ensure that the funders and grant numbers match between the Financial Disclosure field and the Funding Information tab in your submission form. Note that the funders must be provided in the same order in both places as well. Currently, the order of the funders is different in both places.

2) Thank you for stating :"The data that support the findings of this study are publicly available from BioImage Archive with the identifier https://doi.org/10.6019/S-BIAD1242."

Please note that, though access restrictions are acceptable now, your entire minimal dataset will need to be made freely accessible if your manuscript is accepted for publication. This policy applies to all data except where public deposition would breach compliance with the protocol approved by your research ethics board. If you are unable to adhere to our open data policy, please kindly revise your statement to explain your reasoning and we will seek the editor's input on an exemption.

**Reviewers' Comments:**

Reviewer's Responses to Questions

**Part I - Summary**

Reviewer #1: The manuscript has been improved significantly in my opinion. Therefore, I will suggest a minor revision this time.

Reviewer #2: Already described in the original version

**Part II – Major Issues: Key Experiments Required for Acceptance**

Reviewer #1: NO.

Reviewer #2: In this paper I raised three main objections that the authors should address before the publication of the manuscript. Most of these were related to the lack of consistency in some results that made the conclusions based on weak demonstrations, e.g. the RNA label inside the plasmodesma and the non-specificity of the observed label. Most of their answers, if not all, are well argued and they have added some new experiments that clarify the proposed scenario of intercellular transport. The issues related to the specificity of the process (TMV MP) are now better explained in the discussion. They have added the RNA binding-deficient Csy4*[mut]-GFP construct as a control (new S3 Fig), which does not label ‘whorls’ even when the virus is tagged.

**Part III – Minor Issues: Editorial and Data Presentation Modifications**

Reviewer #1: 1. The virus names should not be italic and not capitalized for the first character, e.g., potato virus X but not Potato virus X.

2. line 354, TVCV may not a good example here as I cannot find a figure or description about the localization of VRC to PD in this paper; instead, maybe these two papers (doi: 10.1128/jvi.01898-19; 10.1111/tra.12768)

Reviewer #2: All minor objections have been corrected and/or answered.

PLOS authors have the option to publish the peer review history of their article (what does this mean? ). If published, this will include your full peer review and any attached files.

**Do you want your identity to be public for this peer review?** For information about this choice, including consent withdrawal, please see our Privacy Policy .

Reviewer #1: No

Reviewer #2: No

**Figure resubmission:**
---

## [Editor Report · Decision Letter 2]

17 Mar 2025

Dear TILSNER,

We are pleased to inform you that your manuscript 'Live-cell RNA imaging with the inactivated endonuclease Csy4 enables new insights into plant virus transport through plasmodesmata' has been provisionally accepted for publication in PLOS Pathogens.

Best regards,

Savithramma P. Dinesh-Kumar

Section Editor

PLOS Pathogens

Savithramma Dinesh-Kumar

Section Editor

PLOS Pathogens

Sumita Bhaduri-McIntosh

Editor-in-Chief

PLOS Pathogens

orcid.org/0000-0003-2946-9497

Michael Malim

Editor-in-Chief

PLOS Pathogens

orcid.org/0000-0002-7699-2064
---

## [Editor Report · Acceptance letter]

Dear TILSNER,

We are delighted to inform you that your manuscript, "Live-cell RNA imaging with the inactivated endonuclease Csy4 enables new insights into plant virus transport through plasmodesmata," has been formally accepted for publication in PLOS Pathogens.

Best regards,

Sumita Bhaduri-McIntosh

Editor-in-Chief

PLOS Pathogens

orcid.org/0000-0003-2946-9497

Michael Malim

Editor-in-Chief

PLOS Pathogens

orcid.org/0000-0002-7699-2064